# Deregulated protein homeostasis constrains fetal hematopoietic stem cell pool expansion in Fanconi anemia

Narasaiah Kovuru [1,6], Makiko Mochizuki-Kashio[2,6], Theresa Menna[1], Greer Jeffrey[1], Yuning Hong [3], Young me Yoon[4], Zhe Zhang[5] & Peter Kurre [1] ✉

Demand-adjusted and cell type specific rates of protein synthesis represent an important safeguard for fate and function of long-term hematopoietic stem cells. Here, we identify increased protein synthesis rates in the fetal hematopoietic stem cell pool at the onset of hematopoietic failure in Fanconi Anemia, a prototypical DNA repair disorder that manifests with bone marrow failure. Mechanistically, the accumulation of misfolded proteins in Fancd2$^{-/-}$ fetal liver hematopoietic stem cells converges on endoplasmic reticulum stress, which in turn constrains midgestational expansion. Restoration of protein folding by the chemical chaperone tauroursodeoxycholic acid, a hydrophilic bile salt, prevents accumulation of unfolded proteins and rescues Fancd2$^{-/-}$ fetal liver long-term hematopoietic stem cell numbers. We find that proteostasis deregulation itself is driven by excess sterile inflammatory activity in hematopoietic and stromal cells within the fetal liver, and dampened Type I interferon signaling similarly restores fetal Fancd2$^{-/-}$ long-term hematopoietic stem cells to wild type-equivalent numbers. Our study reveals the origin and pathophysiological trigger that gives rise to Fanconi anemia hematopoietic stem cell pool deficits. More broadly, we show that fetal protein homeostasis serves as a physiological rheostat for hematopoietic stem cell fate and function.

Hematopoietic stem cells (HSCs) are capable of self-renewal, differentiation, and the lifelong production of all blood and immune cells. In order to generate adequate numbers of these multipotent cells during development, HSCs successively colonize different microenvironments[1]. At mid-gestation, the fetal liver (FL) provides critical cues for the rapid expansion of definitive long-term (LT-)HSCs with sufficient regenerative capacity to sustain lifelong organ function. Tonic sterile inflammation is a crucial component for the initial emergence and expansion of the HSC pool during fetal gestation. Type I and -II interferon, as well as transforming growth factor beta (TGF-β) signaling all contribute to pre- and perinatal HSC fate and function[2–4]. To minimize the potential mutational burden in a rapidly proliferating HSC pool, stem and progenitor cells rely on intact DNA repair pathways and cell cycle checkpoints[5,6]. More recent work indicates that a narrowly controlled rate of protein synthesis is another highly conserved feature of somatic stem cells, including LT-HSCs[7].

Fanconi anemia (FA) is a multisystem disorder, caused by mutations in one of 23 FA genes that cooperate in a canonical DNA repair pathway, confer cancer predisposition, and cause bone marrow failure (BMF)[8,9]. The mechanisms that lead to early childhood BMF in FA involve progression through DNA damage-induced cell cycle arrest in HSCs. In mice, this sequence is experimentally replicated by alkylating agent- or aldehyde exposure, but a lack of murine models that phenocopy the prototypical spontaneous BMF phenotype seen in patients has so far precluded a more mechanistic understanding of the

pathophysiology underlying FA-mediated BMF[8,10]. Clinically, rapid postnatal BMF in FA is accelerated by prenatal depletion of the HSC pool, an observation that is extensively documented in very young FA patients as well as fetal tissues and observed in several mouse models[8,11–15]. Prior work by our group and others showed that the prenatal HSPC deficits can be studied in FA mice during unperturbed gestation[12,14,15].

Using this validated model system, we show that the absence of Fancd2 leads to an accumulation of unfolded proteins and an ER stress response that constrains fetal HSC pool formation. We show that these events are triggered by exaggerated sensitivity to fetal sterile inflammation. LT-HSC numbers can be pharmacologically and genetically rescued by restoring protein folding or disrupting inflammation, respectively.

## Results

### A developmental window of vulnerability governs fetal Fancd2$^{-/-}$ HSC defects

A uniquely proliferative fetal HSC phenotype supports the developmental emergence of a finite number of cells that safeguard lifelong blood and immune system function[1,16]. The prenatal depletion in FA precipitates a rapid progression to overt BMF in young children, and occurs spontaneously in several murine models of FA[8,14,15]. To identify the origin of developmental deficits in FA, we first performed detailed immunophenotypic profiling of HSC and progenitor subsets, as previously defined by others[17] and shown in Fig. S1A, in wild-type (WT) and Fancd2$^{-/-}$ mice across ontogeny. Results in FL (E12.5, E13.5, E14.5, and E18.5), fetal BM at E18.5, postnatal BM at P21, and in adult BM at 10- and 30-weeks reveal near equivalent LT-HSC (Lin$^-$/Sca-1$^+$/c-kit$^+$/CD150$^+$/CD48$^-$) number and frequency at E12.5, with significant differences between genotypes first emerging during the HSC expansion from E12.5 to E14.5 (Fig. 1A, B). The data are consistent with studies in Fanca$^{-/-}$ mice where differences in BM LT-HSC frequency become non-significant once HSCs assume the more quiescent adult phenotype[18]. Analysis of myeloid-committed multipotent progenitors (MPP) 2 (Lin$^-$/Sca-1$^+$/c-kit$^+$/CD150$^+$/CD48$^+$) (Fig. 1C) and the lymphoid progenitor enriched MPP3/4 population (Lin$^-$/Sca-1$^+$/c-kit$^+$/CD150$^-$/CD48$^-$) (Fig. 1D) indicate that the emerging Fancd2$^{-/-}$ deficits involve a general decrease in progenitors, whereas the BM MPP2 fraction expands, itself a known proliferative stress response[19]. We reasoned that, if the unprovoked experimental proliferative stress during HSPC expansion in the E13.5 and E14.5 fetal liver contributed to the observed phenotype, then proliferative stress caused by adoptive transfer should reveal similar functional deficits in E12.5 Fancd2$^{-/-}$ HSPCs. Indeed, serial transplantation of E12.5 Fancd2$^{-/-}$ FL cells confirmed compromised repopulation of myeloablated hosts, (Fig. S1B). Together, these results reveal that hematopoietic deficit in Fancd2$^{-/-}$ mice originate during developmental HSC pool expansion.

### An altered FA fetal liver transcriptome at single cell level

To better understand the mechanism that constrains HSPC expansion in the FA fetal liver, we performed single-cell RNA Sequencing of WT and Fancd2$^{-/-}$ FL cells. For adequate representation of smaller HSPC populations, we selectively depleted Ter119 expressing erythroid cells (Fig. 1E). Using Seurat and visualized in UMAP representation, we identify 15 distinct cell clusters (Fig. 1F), based on validated marker gene representation (Fig. S2A), including hematopoietic stem cells (HSCs), multipotent progenitor cells (MPPs), common lymphoid progenitors (CLPs), common myeloid progenitors (CMPs), megakaryocyte and erythroid progenitors (MEMPs), erythroblasts, megakaryocytes, basophils, mast cells, monocyte/DC progenitors, neutrophil progenitors, neutrophils and B-cell progenitors, as well as the small fraction of non-hematopoietic (niche) cell populations (Fig. S2B, C). Notably, none of these clusters is unique to WT or Fancd2$^{-/-}$ genotype, indicating that FL tissue composition itself is not

impacted by the loss of Fancd2. Likewise, lineage differentiation analysis (Stemnet classification) based on normalized lineage-specific gene expression along six key differentiation trajectories (B-cell-, myeloid, DC, mast cell, megakaryocyte, and erythroid cells) revealed no genotype-specific differences (Fig. 1G)[20]. This observation is consistent with previous findings in FA patient BM CD34$^+$ cells[21]. Gene set enrichment analysis (GSEA) of differentially expressed genes on the other hand reveals an upregulation of pathways that govern proteostasis, including "ribosome biogenesis" (NES: 1.8, $p < 0.001$), "translation" (NES: 2.2, $p < 0.001$) and "mTOR signaling" (NES: 1.7, $p < 0.0001$) and MYC target signaling (NES: 2.1, $p < 0.001$) (Fig. 1H).

### FANCD2 loss causes delayed S-phase entry through replication stress in FL HSC

While the transcriptome analysis with MYC target and ribosome biogenesis pathway enrichment suggested strong proliferative pressure in Fancd2$^{-/-}$ littermate HSPCs, there was a clear deficit in expansion in the fetal HSC pool. To reconcile these seemingly conflicting observations, we next examined HSPC cell cycle activity by timed 5-ethyl-2'-deoxiuridine/ Bromodeoxyuridine (EdU/BrdU) sequential injection into pregnant E13.5 dams, whose BM HSPCs serve as a slower cycling adult phenotype control population[22]. Conceptually, cells entering S-phase during the two hours following EdU injection will initially become EdU$^+$. This is followed by BrdU injection, when cells newly entering S-phase become EdU$^-$/BrdU$^+$, cells exiting S-phase become EdU$^+$/BrdU$^-$, and those that continue to remain in S-phase stain double positive EdU$^+$/BrdU$^+$ (Fig. 2A). Our results show a significantly greater fraction of Fancd2$^{-/-}$ FL HSC and HSPC stain EdU$^+$ and/or BrdU$^+$ compared to WT, indicating an increased population of cells containing newly replicating single strand DNA (ssDNA), (Fig. 2B). Concurrently, the frequency of EdU$^-$/BrdU$^+$ LT-HSC and Lin$^-$ cells, indicative of delayed S-phase entry, was decreased in Fancd2$^{-/-}$ compared to WT (Fig. 2C). During S-phase, FANCD2 cooperates in sensing and resolving experimental replication stress[23,24]. We therefore hypothesized that its physiologic role in the fetal HSPC pool is to counter replication stress. Our data in Fancd2$^{-/-}$ HSPC confirm increased phosphorylation of Replication-associated protein (RPA)32, which stabilizes nuclear ssDNA during stalled replication (Fig. 2D). We also observed significantly increased phosphorylation in mini chromosome maintenance (MCM) 2 at the direct ATR substrate target site (Ser108), a known marker of replication fork stalling associated with replication stress and cell cycle defects, in Fancd2$^{-/-}$ fetal HSPCs[25] (Fig. 2E) Phosphorylation of RPA-ATR is typically followed by phosphorylation of Chk1 (pChk1), and we observed a significant increase in pChk1 immunostaining in Fancd2$^{-/-}$ HSPC (Fig. 2F). By contrast, the E12.5 FL HSPCs before expansion deficits are seen and quiescent postnatal HSPCs from 8-week-old animals fail to show significant differences in pChk1 between WT and Fancd2$^{-/-}$ littermates (Fig. 2G). Together, these results confirm a fetal pathophysiologic vulnerability, whereby Fancd2$^{-/-}$ FL HSPC experience MYC activation, revealing for the first time a physiologic role for FANCD2 in fetal replication stress response.

### Increased protein synthesis and proteostatic stress in Fancd2$^{-/-}$ FL HSPCs

As a functional corollary, we anticipated that increased ribosome biogenesis as a GSEA signature in Fancd2$^{-/-}$ FL HSPC will lead to raised protein synthesis rates[26–28]. We therefore set out to measure the incorporation of the puromycin analog O-Propargyl-puromycin (OPP) in HSPC in vivo (Fig. 3A). First, we confirmed the relatively increased OPP staining in fetal- compared to adult phenotype HSPCs from the maternal BM[3]. Between genotypes, the fetal Fancd2$^{-/-}$ HSPC compartment revealed gains in protein synthesis across all subpopulations that reach significance in LT-HSCs (Fig. 3B). Reports show that HSCs and other somatic stem cells critically depend on low protein synthesis to match cell specific folding capacity, whereby even small,

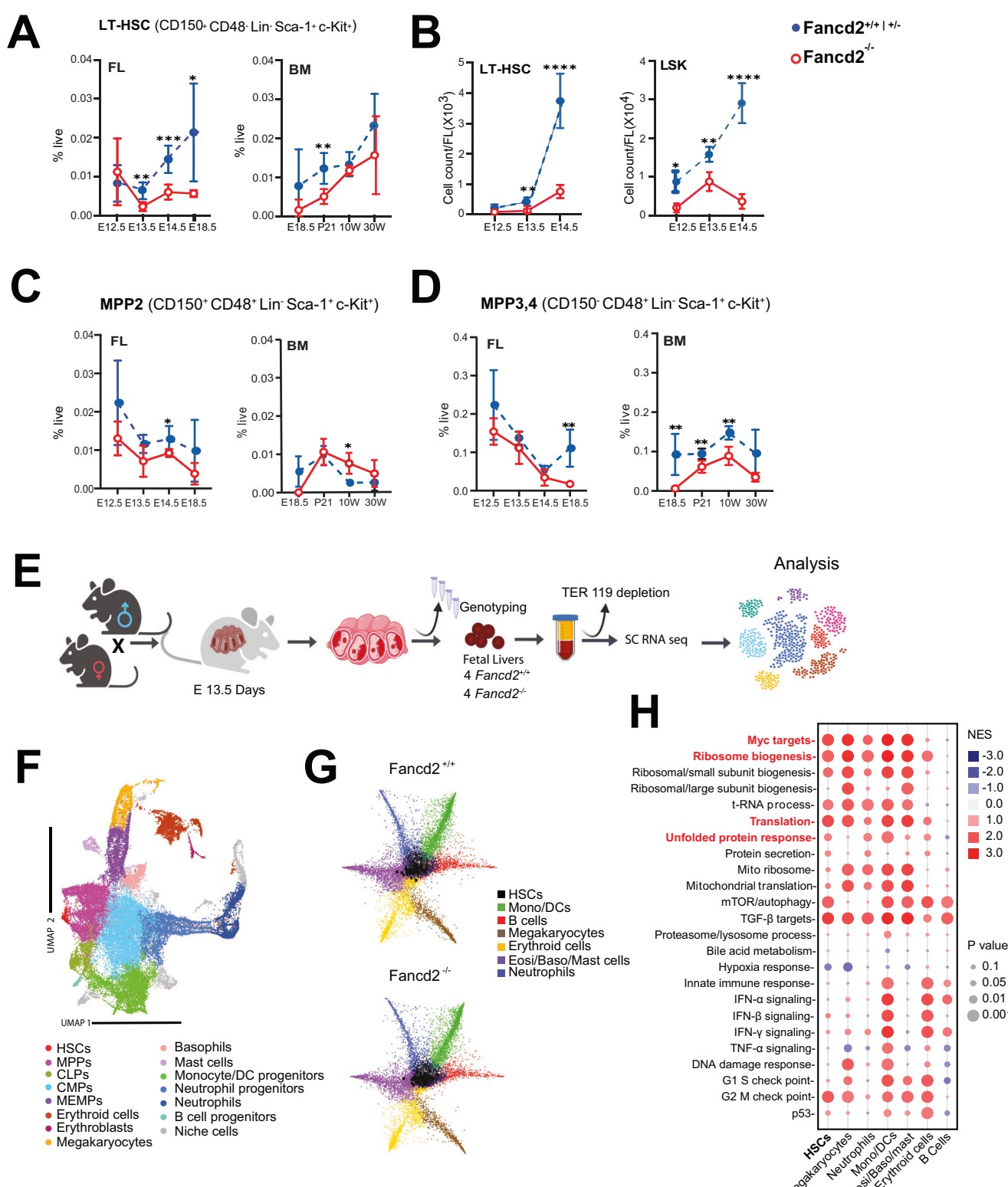

experimentally induced gains in protein synthesis functionally compromise HSC function through accumulation of unfolded proteins[7]. Tetraphenylethene maleimide (TPE-MI) is a cell-permeable dye that fluoresces upon binding to free thiol side chains of cysteine molecules characteristically exposed in unfolded proteins[29]. We confirm that experimental induction of mis/unfolded proteins with thapsigargin, an inhibitor of ER-calcium ATPase, leads to increased cellular TPE-MI fluorescence (Fig. S3A). Next, in fetal Fancd2[−/−] HSPCs from experimentally unperturbed timed pregnancies, the TPE-MI assay reveals

excess unfolded proteins across all Fancd2[−/−] subsets, including LT-HSCs, compared with WT (Fig. 3C). This observation also aligns with our GSEA analysis (Fig. 1H), which showed enrichment of unfolded protein response genes in FL Fancd2[−/−] HSPCs. To rule out differences in the degradation of unfolded proteins in FL Fancd2[−/−] HSPCs, we also analyzed proteasome activity in LSK cells using the proteasome-Glow chymotrypsin like cell-based assay (Promega), demonstrating balanced proteasome activity between genotypes (Fig. 3D). Representative immunofluorescent images of cells stained with the

**Fig. 1 | Deficits in expansion and prolonged S-phase transition in Fancd2⁻/⁻ FL HSC. A** Immunophenotyping was performed to determine the frequency of LT-HSC: CD150⁺ CD48⁻ Lin⁻ Sca-1⁺ c-Kit⁺ (LSK), in WT and Fancd2⁻/⁻ littermates across indicated time points in ontogeny. **B** Absolute numbers of LT-HSC (left panel) and LSK (right panel) in E12.5-13.5-14.5 of WT (⁺/⁺) compared with Fancd2⁻/⁻ FL. **C** Frequency of MPP2 (CD150⁺ CD48⁺ Lin⁻ Sca-1⁺ c-Kit⁺). **D** Frequency of MPP3,4 (CD150⁻ CD48⁺ Lin⁻ Sca-1⁺ c-Kit⁺). (E12.5: WT $n = 4/3$, Fancd2⁻/⁻ $n = 7/3$; E13.5: WT $n = 9/4$, Fancd2⁻/⁻ $n = 5/4$; E14.5: WT $n = 9/4$, Fancd2⁻/⁻ $n = 6/4$; E18.5: WT $n = 7/3$, Fancd2⁻/⁻ $n = 3/3$; P21 WT $n = 5/3$, Fancd2⁻/⁻ $n = 6/3$, 10 Weeks (10 W): WT $n = 4/3$, Fancd2⁻/⁻ $n = 4/3$; 30 W: WT $n = 6/3$, Fancd2⁻/⁻ $n = 5/3$. Data in (**A–D**) are represented ±SEM. Welch's $t$-test was used for statistical analysis (*$P < 0.05$, **$P < 0.01$, ***$P < 0.001$, ****$P < 0.0001$). The statistical analyses in (**A–D**) are two sided and were performed with GraphPad Prism7.0 software. **E** Schematic diagram of the procedures for fetal liver (WT and Fancd2⁻/⁻) capture, processing, single cell-RNA seq profiling, and analysis. Schematic figure was created with BioRender.com released under a Creative Commons Attribution-Noncommercial-NoDerivs 4.0 International license. **F** UMAP visualization of the fetal liver cell composition. Schematic (**G**) Lineage trajectory analysis of HSPCs with progression of immature HSPCs at the center toward fully differentiated cells (neutrophils, megakaryocytes, erythrocyte progenitors, eosinophil/basophil/mast cells, monocytes/dendritic cells and B-cells) at the end of each projection. Left hand panel: WT, right hand panel: Fancd2⁻/⁻. **H** Dot plot representation of the normalized enrichment score- (NES) and $P$-values of Gene Set Enrichment Analysis (GSEA) in Fancd2⁻/⁻ versus WT HSPCs, NES and $P$ values of **H**. GSEA analysis contains statistical outputs from the GSEA software. Source data are provided as a Source Data file.

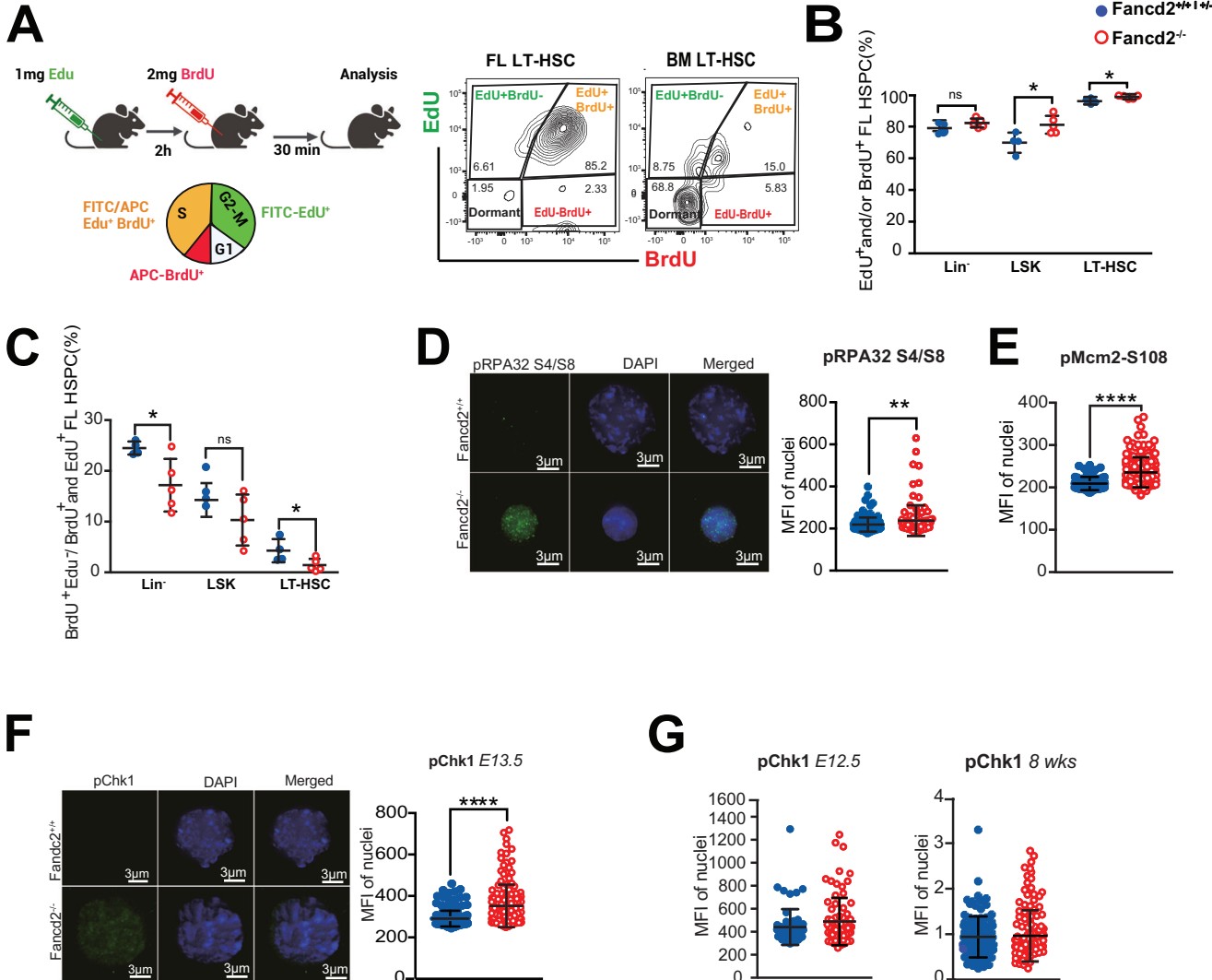

**Fig. 2 | Fancd2⁻/⁻ fetal liver HSPCs show replication stress and delayed cell cycle progression. A** Schema for sequential EdU/BrdU injection in the dam at E13.5 with cell cycle analysis, Representative flow panels illustrate FL (left) and BM (right) HSPC distribution with predicted differences in dormancy, lower left quadrant. Schematic Figure was created with BioRender.com released under a Creative Commons Attribution-NonCommercial-NoDerivs 4.0 International license. **B** Frequency of nascent ssDNA (EdU⁺ and/or BrdU⁺) in the Lin-, LSK and LT-HSC: (Fancd2⁺⁺ $n = 4$, Fancd2⁻/⁻ $n = 5$); *$P < 0.05$. **C** Percentage of cells newly entering S-phase (BrdU⁺EdU⁻ per EdU⁺ and/or BrdU⁺) in Lin-, LSK and LT-HSC: WT (Fancd2⁺⁺) $n = 4$, Fancd2⁻/⁻ $n = 5$). **D** Representative images and Immunofluorescence (IF) analysis of WT (Fancd2⁺⁺) and Fancd2⁻/⁻ HSPC pRPA32 S4/S8 (E13.5; WT: $n = 5$; 116 cells, Fancd2⁻/⁻: $n = 4$; 111 cells). **E** Immunofluorescence (IF) analysis of Fancd2⁺⁺ and Fancd2⁻/⁻ HSPC pMCM2-S108 (E13.5; WT(Fancd2⁺⁺) $n = 4$ pups, 64 cells, Fancd2⁻/⁻: $n = 6$ pups, 149 cells). **F** pChk1-S345 WT(Fancd2⁺⁺): $n = 9$; 174 cells, Fancd2⁻/⁻: $n = 4$; 145 cells at E13.5 days). **G** Representative images and IF analysis pChk1-S345 at E12.5 WT (Fancd2⁺⁺) $n = 2$: 63 cells; Fancd2⁻/⁻: $n = 5$: 81 cells) and in 8-week-old adult BM (WT $n = 2$: 106 cells; Fancd2⁻/⁻: $n = 2$: 113 cells). Welch's $t$-test was used for statistical analysis, Data is represented ±SEM, all the statistical analysis are two sided. *$P < 0.05$, **$P < 0.01$, ***$P < 0.001$, ****$P < 0.0001$. All the statistical analysis are two sided and were performed with GraphPad Prism7.0 software. Source data are provided as a Source Data file.

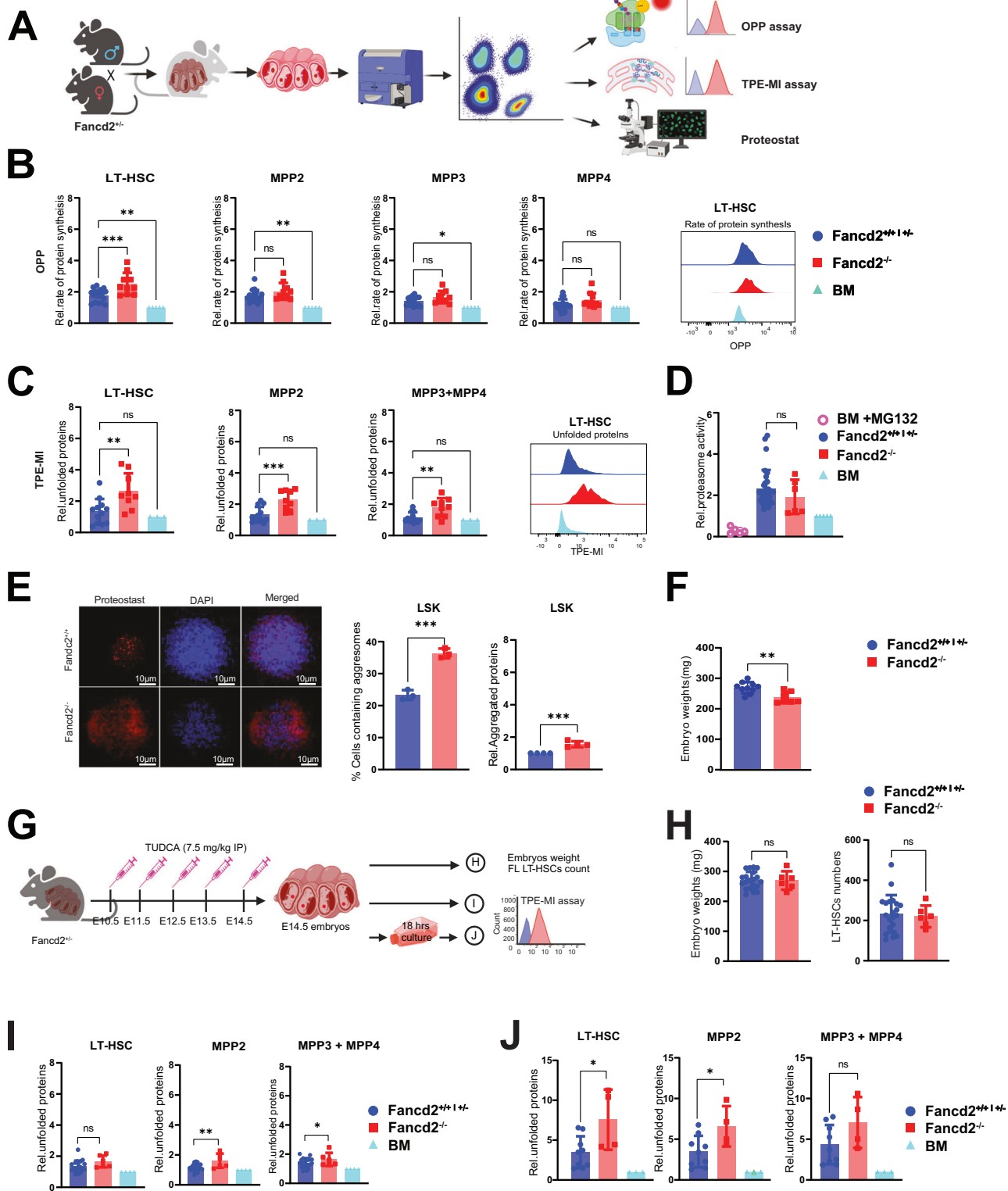

Proteostat™ agent and their quantification show the anticipated increase in the frequency of cells with elevated perinuclear aggregates in Fancd2⁻/⁻ cells and the relative amount of perinuclear aggregates (Fig. 3E). In addition, we found that embryo weights were decreased in Fancd2⁻/⁻, consistent with FA as multisystemic developmental disorder (Fig. 3F). To ameliorate the accumulation of unfolded proteins in Fancd2⁻/⁻ FL HSPCs we used the bile acid tauroursodeoxy cholic acid (TUDCA), a chemical chaperone reported to improve protein folding and proliferation[30,31]. We injected TUDCA daily in timed pregnancies

from E10.5-E14.5 and studied littermates shortly following harvest after the last dose (Fig. 3G). TUDCA supplementation rescued embryo size (by weight), along with LT-HSC numbers (Fig. 3H), and intracellular staining of unfolded proteins by TPE-MI demonstrated that it also prevented the accumulation of unfolded proteins in Fancd2⁻/⁻ FL LT-HSCs and (Fig. 3I). Representative flow cytometry plots for LT-HSCs gating are shown in Fig. S5A. Predictably for a transient pharmacological treatment, Fancd2⁻/⁻ FL HSPCs from the TUDCA protected pregnancies, showed increased levels of unfolded proteins when

**Fig. 3 | Dysregulated protein homeostasis in Fancd2$^{-/-}$ fetal liver HSPCs.**
**A** Schematic representation of experimental procedures. Schematic Figure was created with BioRender.com released under a Creative Commons Attribution-NonCommercial-NoDerivs 4.0 International license. **B** Analysis of the rate of translation by in vivo OPP incorporation in Fancd2$^{+/+}$ and Fancd2$^{+/-}$ (FL Samples/Pregnancies $n = 18/5$), Fancd2$^{-/-}$ (FL Samples/Pregnancies $n = 10/5$), normalized to the dam BM (Fancd2$^{+/-}$: $n = 5$ pregnant mice) LT-HSCs ($P = 0.0007$), MPP2 ($P = 0.03$), MPP3 ($P = 0.06$) and MPP4 ($P = 0.1$). Representative histograms from an analysis of LT-HSC. **C** Analysis of unfolded protein levels by TPE-MI analysis in Fancd2$^{+/+}$ and Fancd2$^{+/-}$ (FL Samples/Pregnancies $n = 12/3$) and Fancd2$^{-/-}$ (FL Samples/Pregnancies $n = 9/3$) FL relative to the Fancd2$^{+/-}$ dam BM ($n = 3$ pregnant mice) LT- HSCs ($P = 0.007$), MPP2 ($P = 0.0009$) and MPP3 + MPP4 ($P = 0.007$). Representative histograms from a LT-HSC analysis. **D** Proteasome activity in LSK cells from Fancd2$^{++}$, Fancd2$^{+/-}$ ($n = 17/5$) and Fancd2$^{-/-}$ ($n = 6/5$) ($P = 06$) FL relative to the Fancd2$^{+/-}$ dam BM ($n = 5$). **E** Representative images of protein aggregates in FL HSPC (Lin$^{-ve}$, Sca-1$^+$ and c-Kit$^+$ cells) from Fancd2$^{+/+}$ ($n = 3/2$), and Fancd2$^{-/-}$ ($n = 3/2$) littermates ($P = 0.0002$). Quantification of relative protein aggregates and percent of aggresome positive cells. **F** Fancd2$^{+/+}$ ($n = 10/4$) and Fancd2$^{-/-}$ ($n = 8/4$) littermate embryo weights, ($P = 0.002$). **G** Experimental design of TUDCA injection. Pregnant mice were intraperitoneally injected with 7.5 mg/kg of TUDCA daily from E10.5 to E14.5. Fetal livers were harvested 2 h after the last injection, for analysis. Schematic of experimental design was created with BioRender.com released under a Creative Commons Attribution-NonCommercial-NoDerivs 4.0 International license. **H** Embryo weights ($P = 0.1$) and LT-HSC numbers ($P = 0.2$) from littermates of TUDCA injected Fancd2$^{+/+}$ and Fancd2$^{+/-}$ ($n = 19/4$) and Fancd2$^{-/-}$ ($n = 6/4$). **I** Analysis of unfolded proteins from TUDCA injected Fancd2$^{+/+}$ and Fancd2$^{+/-}$ ($n = 19/4$), and Fancd2$^{-/-}$ ($n = 6/4$) fetal liver LT-HSCs ($P = 0.2$), MPP2 ($P = 0.003$), MPP3 + MPP4 ($P = 0.2$) normalized to respective Fancd2$^{+/-}$ maternal BM (right-most column) ($n = 4$). **J** Unfolded protein analysis in FL cells from a TUDCA-injected pregnant dam followed by sacrifice, BM harvest, and subsequent ex vivo culture for 18 h without TUDCA Fancd2$^{+-}$ and Fancd2$^{+/-}$ ($n = 9/3$) and Fancd2$^{-/-}$ ($n = 4/3$). Data are represented as mean ± SEM. In **B**, **C**, **D**, **H** and **J** one-way ANOVA was considered for statistical analysis and for panel -**E**, -**F** and -**I**, $t$-test was used for statistical analysis. All the statistical analysis are two sided and were performed with GraphPad Prism7.0 software. *$P < 0.05$, **$P < 0.01$, ***$P < 0.001$, and ns: non-significant. Source data are provided as a "Source Data" file.

subsequently cultured ex vivo in the absence of TUDCA (Fig. 3J, Fig. S3E–H).

To test the impact of deregulated proteostasis during proliferative stress on the functional fitness of Fancd2$^{-/-}$ HSPCs. We performed competitive transplantation of equal numbers of Fancd2$^{+/+}$ (CD45.1/45.2) and Fancd2$^{-/-}$ (CD45.2) whole BM cells into irradiated C57BL/6 (CD45.1) recipient mice. This was followed by IP injection of TUDCA (7.5 mg/kg) or diluent in recipients, daily for one week. Serial analysis after transplantation showed a modest, but significant transient increase in peripheral blood chimerism from TUDCA exposed Fancd2$^{-/-}$ cells, consistent with the role of a chaperone that requires durable supplementation (Fig. S4C). To explore the possibility of proteostasis defects in FA patients, we mined previously published RNA sequencing data from healthy and FA patient HSPCs[21]. Strikingly, FA patient HSPCs showed significantly elevated levels of unfolded protein response and endoplasmic reticulum stress-associated degradation (ERAD) genes (Fig. S4D–E). Altogether, these results indicate that both Fancd2$^{-/-}$ murine FL HSPCs and human FA patient HSPCs have deregulated proteostasis.

## FANCD2 restrains inflammation and MYC expression under experimental replication stress

Given the strong transcriptomic GSEA signature of MYC targets, we used intracellular flow-cytometry to confirm MYC protein levels in all HSPC subpopulations, with a significant increase in Fancd2$^{-/-}$ FL LT-HSCs compared to Fancd2$^{+/+}$ fetal liver and maternal (Fancd2$^{+/-}$ dam) bone marrow cells (Fig. 4A).

MYC functions as a transcription factor and we further validated the increased activity of MYC target expression by multiplex RT$^2$ profiler array. Results reveal elevated expression of multiple ribosomal protein genes such as RPL5, RPL23, RPL13, activating transcription factor 4 (ATF4) a component of the unfolded protein response (all consistent with dysregulated proteostasis) as well as other canonical targets in Fancd2$^{-/-}$ FL-HSPCs (Fig. S5B).

We wished to test if direct MYC inhibition could rebalance proteostasis to rescue FL HSPC number and function using (+)-JQ1, a BRD4 super-enhancer inhibitor, that suppresses MYC transcription. However, in a series of studies exploring IP injection of different doses in the dam, JQ1 treatment invariably led to fetal demise. MYC protein levels in quiescent, non-proliferating BM Fancd2$^{-/-}$ HSPCs (Fig. S5C) and in MEFs not under replication stress (Fig. 4G) as well as the rate of protein synthesis remain unchanged compared to adult BM Fancd2$^{+/+}$ cells (Fig. S5D), but this changes under proliferative stress conditions. In a validated polyvinyl ETOH supplemented ex vivo culture, FA HSPCs lag behind the Fancd2$^{+/+}$ control cells in expansion, and experience significantly elevated levels of unfolded proteins (Fig. S4A, B)[32,33]. Here,

adult BM Fancd2$^{-/-}$ HSPCs show increased c-MYC activity (Fig. S5E). However, when we treated the ex vivo expanded Fancd2$^{-/-}$ BM LT-HSCs with (+)-JQ1 (as a surrogate for proteostasis-challenged fetal HSPC), MYC inhibition resulted in a profound loss of HSPC viability in both Fancd2$^{+/+}$ and Fancd2$^{-/-}$ progenitor survival, wherein Fancd2$^{-/-}$ growth in particular was so severely and disproportionately restricted to preclude meaningful downstream analysis (Fig. S5F). These results support MYC as a critical component in maintaining fetal development and proliferating Fancd2$^{-/-}$ HSPCs.

To understand the potential trigger for both increased MYC signaling and proteostatic deregulation in fetal Fancd2$^{-/-}$ FL HSPCs we considered that MYC protein levels in adult WT HSPCs are regulated by IFN-α inflammatory signals[34]. Fancd2$^{-/-}$ HSPCs are vulnerable to inflammation, and serial experimental injection of poly-(I:C) leads to increased MYC levels in adult FA HSPC[21]. We therefore reanalyzed our transcriptome data for an inflammatory signature in fetal liver hematopoietic and non-hematopoietic fetal liver niche cells and observed increased activity in key inflammatory pathways, such as Type-1- (NES: 1.4, $p = 0.08$), Type-2 IFN signaling (NES: 1.2, $p = 0.16$) and TGF-β (NES: 2.3, $p < 0.001$) in select HSPC subsets (Fig. 1H). Specific analysis of FL niche cells showed six discrete cell types, including hepatocytes, endothelial cells, cholangiocytes, mesenchymal cells, hepatic stellate cells, with two distinct populations of Kupffer cells (M1 and M2) (Fig. S2B, C). Remarkably, this analysis in Fancd2$^{-/-}$ FL niche cells also shows broad enrichment for inflammatory pathways Fancd2$^{-/-}$ mesenchymal cells shows enriched interferon-γ (NES:1.5, $p = 0.05$) and interleukin-4 (NES:1.8, $p = 0.001$), Fancd2$^{-/-}$ hepatoblasts are enriched for innate immune response pathway (NES:1.3, $p = 0.002$) and response to IL-4 (NES:1.7, $p = 0.004$) (Fig. S2D), with no difference in differentiation trajectory analysis of Fancd2$^{-/-}$ mesenchymal cells between the two genotypes (Fig. S2E), while Fancd2$^{-/-}$ hepatoblasts show skewed differentiation towards hepatocytes, and a reduced cholangiocyte population (Fig. S2F). We were particularly interested in the inflammatory Kupffer cells where we observed polarization toward two distinct populations of tissue-resident macrophages (Kupffer cells; M1 vs M2). (Fig. 4B). While the M1/M2 frequencies between genotypes remain unchanged, GSEA analysis showed enriched for interferon-α pathway responses (M1: NES:2.3, M2: NES:1.4 (Fig. 4C). Fancd2$^{-/-}$ Kupffer cell subtypes are also enriched for MYC targets M1: NES:2.3, M2; NES:2.0) (Fig. 4C) Thus, gene ontology analysis of hematopoietic- as well as niche cells from Fancd2$^{-/-}$ littermates reveals enrichment for several inflammatory pathways.

Both replication stress and inflammatory stimulation can increase MYC levels[34,35], and experimental inflammation triggers a MYC response in adult FA BM HSPC. From a pathophysiological perspective, we reasoned that the constitutive loss of FANCD2 may sensitize the

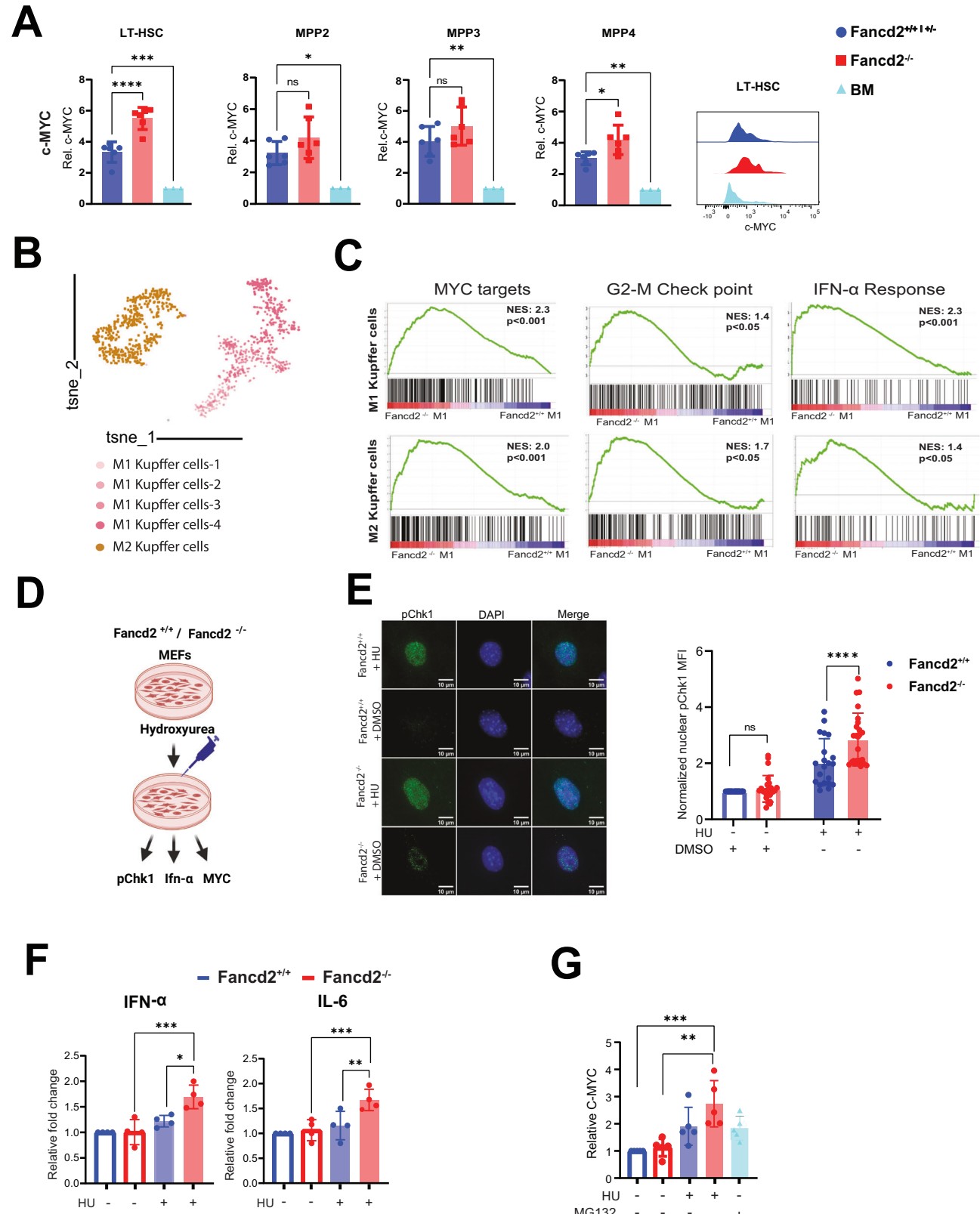

fetal HSPC pool to sterile inflammatory cues normally required for its developmental formation, thereby creating a potential trigger for MYC and proteostatic stress. To test this sequence of events, we generated Mouse Embryonic Fibroblasts (MEFs) from Fancd2[+/+] and Fancd2[-/-] fetuses and treated these MEFs at passage 2–3 with hydroxyurea (HU) to experimentally induce replication stress (Fig. 4D). This elicits the

predicted replication stress response involving phosphorylation of p-Chk1 in Fancd2[-/-] MEFs by immunofluorescence analysis (Fig. 4E). Using the identical experimental set-up, we also observe the anticipated increase in IFN-α and IL-6 gene expression in Fancd2[-/-] MEFs (Fig. 4F), at the transcript levels using the target gene primers listed in supplementary Table 3. We further confirmed that HU-mediated

**Fig. 4 | Experimental replication stress induces inflammatory cytokines and MYC expression in Fancd2⁻/⁻ MEFs. A** Relative increase in MYC levels in Fancd2⁻/⁻ fetal liver LT-HSCs ($P = 0.0001$), MPP2 ($P = 0.2$), MPP3 ($P = 0.01$) and MPP4 (Fancd2⁺/⁺: $n = 5/3$, Fancd2⁻/⁻: $n = 6/3$), normalized to the respective dam BM ($n = 3$) representative histograms from a LT-HSCs. **B** t-SNE plot showing the Kupffer cells subtypes (M1, M2) and their differentiation trajectory. **C** Gene set enrichment analysis demonstrating enrichment of MYC targets, G2-M check point, IFN-α response in Fancd2⁻/⁻ M1 Kupffer cells (upper panel) and Fancd2⁻/⁻ M2 Kupffer cells (lower panel), both NES and *P* values are statistical outputs from the GSEA software analysis. **D** Experimental plan of replication stress followed by inflammatory cytokines and endogenous c-MYC analysis in MEFs. Schematic of experimental plan was created with BioRender.com released under a Creative Commons Attribution-NonCommercial-NoDerivs 4.0 International license. **E** Representative immuno-fluorescence (IF) images and quantification of pChk1-S345 in Fancd2⁺/⁺ and Fancd2⁻/⁻ MEFs at basal conditions and under replication stress using hydroxyurea

(HU) at 200 μM final concentration ($n = 3$ independent experiments, DMSO +Fancd2⁺/⁺ vs DMSO+Fancd2⁻/⁻ MEFs $P = 0.3$). HU+Fancd2⁻/⁻ vs HU+Fancd2⁻/⁻ MEFs $P = 0.0008$). **F** Real time PCR analysis of inflammatory cytokines IFN-α (Basel Fancd2⁻/⁻ vs Fancd2⁻/⁻ + HU $P = 0.0007$), Fancd2⁺/⁺ + HU vs Fancd2⁻/⁻ + HU, ($P$ value = 0.01), and IL-6 Basel KO vs KO + HU $P = 0.006$), Fancd2⁺/⁺+HU vs Fancd2⁻/⁻ + HU, ($P$ value = 0.01) (n = 4 independent experiments). **G** Flow cytometric analysis of MYC levels in Fancd2⁺/⁺ and Fancd2⁻/⁻ MEFs under at both basal conditions and replication stress (HU: 200 μM working concentration) conditions, using proteasome inhibitor MG132 (2 μM final concentration) as a positive control ($n = 5$ independent experiments, basal Fancd2⁻/⁻ vs Fancd2⁻/⁻ + HU, $P$ value = 0.001), basal Fancd2⁺/⁺ vs Fancd2⁻/⁻ + HU, $P$ value = 0.0006). Data are represented as mean ± SEM. In **A**, **E**, **F**, and **G**. and one-way ANOVA was considered for statistical analysis, *$P < 0.05$, **$P < 0.01$, ***$P < 0.001$, ****$P < 0.0001$, and ns: non-significant. All statistical analyses are two sided and were performed with GraphPad Prism7.0 software. Source data are provided as a Source Data file.

replication stress indeed raises MYC at the protein level in MEFs. As a positive control we show that MG132 (proteasome inhibitor, that slows MYC degradation) raises MYC levels[7] (Fig. 4G). Together with our single-cell transcriptome observations in stromal and hematopoietic components of Fancd2⁻/⁻ fetal livers, the MEF experiments support a model whereby replication stress in Fancd2⁻/⁻ fetuses adds to physiologic sterile inflammatory signals, raising MYC levels and protein synthesis to rates that exceed the tightly regulated fetal HSPC folding capacity. We next set out to test this sequence directly in vivo.

### Ifnar1 haploinsufficiency partially rescues Fancd2⁻/⁻ HSPC defects

Adult FA patients show a baseline increase in proinflammatory cytokines in the BM plasma and active MYC signaling in BM HSPC. Experimentally, injection of poly(I:C) in adult mice induces IFN-α signaling and activates MYC signaling[34]. Sterile inflammation is critical for the fetal hematopoietic development, but the fetal environment in Fancd2⁻/⁻ mice reveals evidence of exaggerated inflammatory activity. We reasoned that a blunted inflammatory response would ameliorate the disruption in proteostasis and safeguard expansion of Fancd2⁻/⁻ FL HSPCs. We therefore generated murine crosses of interferon alpha receptor chain 1 (Ifnar1) and FANCD2 (Fancd2⁻/⁻) deficiency. Initial experiments revealed that the concurrent biallelic loss of both Ifnar1 and Fancd2 resulted in a sharp decline in fetal liver cells (Fig. S3B), a decreased HSPC pool (Fig. S3C) and small, diffusely abnormal fetal livers (Fig. S3D). Incidentally, this is quite similar to the loss of viability and compromised HSPC pool seen with complete loss of TGF-β signaling[36]. Therefore, to maintain some level of fetal sterile inflammation required for normal HSPC development, we adopted a breeding schema that generated littermates that were heterozygous (⁻/⁺) for Ifnar1 and either Fancd2⁻/⁻ or Fancd2⁺/⁺ (WT). When we analyzed fetal liver HSPCs from the Fancd2⁻/⁻/Ifnar1⁺/⁻ versus Fancd2⁺/⁺/Ifnar1⁺/⁻ or Fancd2⁺/⁻/Ifnar1⁺/⁻ offspring (Fig. 5A), we observed that heterozygous loss of the Ifnar1 allele partially restores physiologic (lower) MYC levels in Fancd2⁻/⁻ fetal liver HSPCs, including in the LT-HSCs (Fig. 5B). To determine a potential fetal FA phenotype rescue, we then systematically evaluated the key functional metrics of the fetal HSPC deficits Fancd2⁻/⁻ established so far. Here, the loss of one Ifnar1 allele indeed partially restored the rate of translation (OPP incorporation) in Fancd2⁻/⁻ compared to Fancd2⁺/⁺ in Ifnar1- haploinsufficient littermates (Fig. 5C). We further evaluated unfolded protein levels using the TPE-MI assay and results again show normalization to WT levels in the combined Ifnar⁺/⁻ Fancd2⁻/⁻ genotype (Fig. 5D). Critically, phenotypic correction through loss of a single Ifnar1 allele was accompanied by rescue of fetal liver cellularity and restored HSPC numbers in Fancd2⁻/⁻ littermates to WT levels (Fig. 5E). Representative flow cytometry plots for these experiments are shown in Fig. S5A. To determine the functional impact, we plated Ifnar⁺/⁻ Fancd2⁻/⁻ FL HSPC for colony forming unit (CFU) assay and showed that Ifnar⁺/⁻ Fancd2⁻/⁻ FL progenitors

provide significantly improved colony formation compared with Fancd2⁻/⁻ fetal liver cells (Fig. 5F). The same observation holds true for adult Ifnar1⁺/⁻ Fancd2⁻/⁻ BM cells (Fig. 5F). To explore alterations of sterile inflammation as a potential contributor to human Fanconi anemia pathogenesis we also mined human FA RNA sequencing data[21] and found significantly elevated levels of IFN-α receptor-1(IFNAR1), IFN-α receptor-2 (IFNAR2) and IFN-γ receptor-2 (IFNGR2) genes in human FA patient HSPCs (Fig. S5E). Altogether, our data support a novel role for FANCD2 in restraining sterile inflammation and protein synthesis to maintain proteostasis and safeguard FL HSC pool expansion (Fig. 5G).

## Discussion

Here, we report the pathophysiologic origin of HSC deficits in Fancd2⁻/⁻ fetuses. The mechanism by which this occurs involves an ER stress response that disrupts protein homeostasis and constrains developmental HSC pool expansion. Mechanistically, our study points to the potentially deleterious consequences of inflammation, otherwise required for emergence and expansion of the developing fetal HSC pool and suggests a strategy for pharmacologic mitigation of proteostasis disruption in FA[37,38].

The fetal HSC phenotype is highly proliferative, supporting the emergence and coordinate expansion of the fetal HSC pool that sustains lifelong hematopoietic and immune function. Crosstalk with stromal and other components in the successive niches of fetal liver and BM continuously adapts HSC output to physiological needs, a system guarded by genome repair and balanced protein metabolism. FA-associated BMF is generally thought to reflect the role of the canonical FA pathway in DNA repair. In patients and murine models, the FA HSC pool is already sharply reduced at birth, precipitating symptomatic BMF early in the life of patients[8,15,21]. However, the endogenous triggers for FA BMF in general, and fetal losses in particular, remain to be clarified.

To begin resolving the origins of hematopoietic failure in FA we first narrowed the emergence of HSC pool deficits in a murine Fancd2⁻/⁻ model to mid-gestation. Experiments show that these deficits in expansion coincide with a pathophysiological p-Chk1 response, evident between E12.5 and 14.5, when delayed S-phase progression restrains the otherwise rapid expansion among Fancd2⁻/⁻ fetal liver HSPCs[23,39,40]. The data reinforce the known role of p-Chk1 in maintaining HSC pool integrity under proliferative stress and illustrate that fetal FA HSPC deficits, unlike the adult FA phenotype, result from a lack of expansion, not apoptotic attrition[6].

To better understand the molecular events that account for the observed phenotypic deficits in the fetal Fancd2⁻/⁻ HSC pool we turned to single cell RNA sequencing data, where GSEA analysis of Fancd2⁻/⁻ FL HSPC pointed to deregulated proteostasis, with strong induction of MYC targets, enhanced ribosome biogenesis and ER stress. With translation as the most error prone aspect of gene expression,

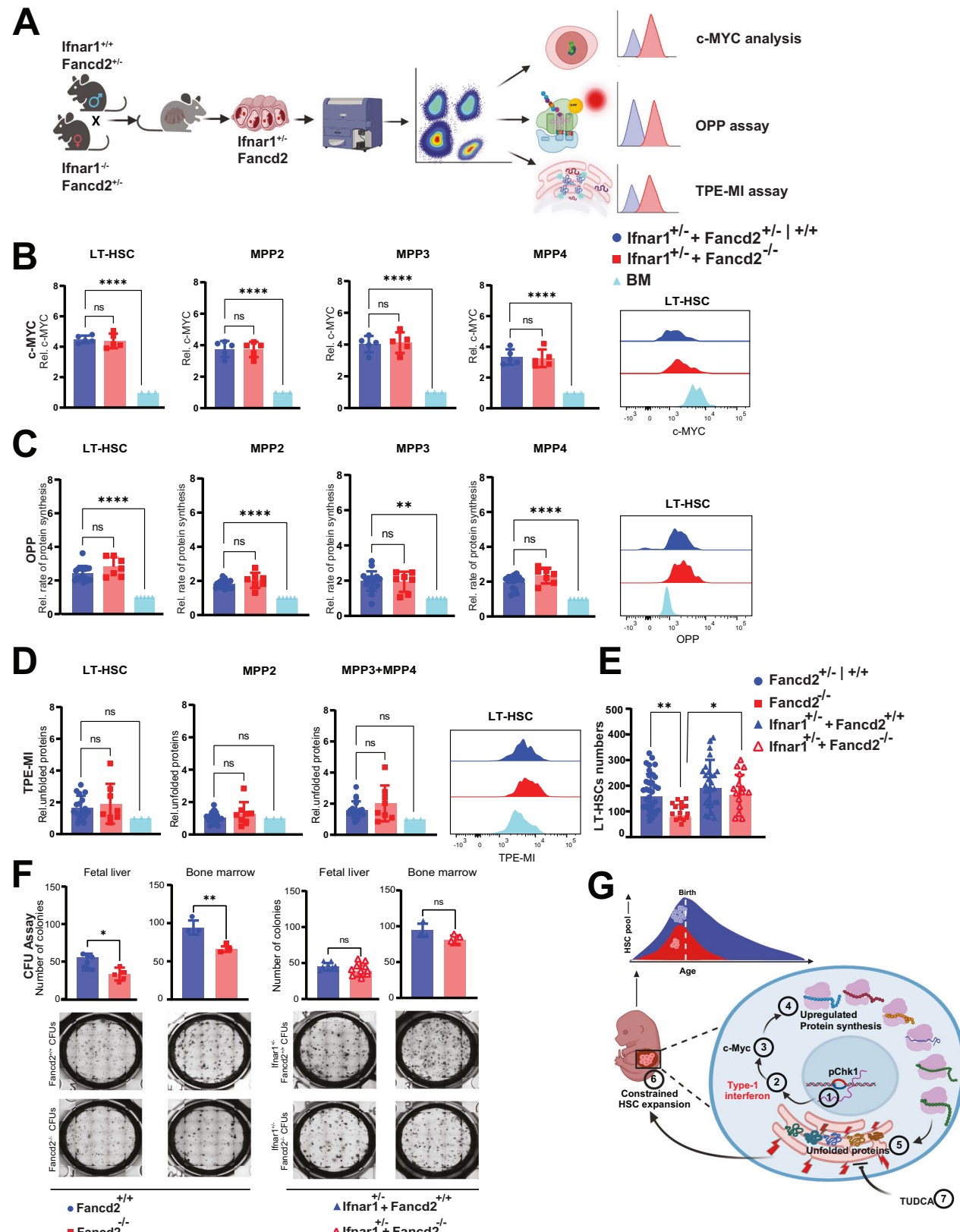

controlled rates of protein synthesis are critical for HSC integrity[7] in particular. Disruption of proteostasis with loss of function, seen in heritable "ribosomopathies" such as Diamond Blackfan-Anemia or Shwachman-Diamond Syndrome are well understood to cause BMF[41–43]. But similarly, gains in ribosomal protein transcription and synthesis rates during aging or following experimental PTEN

inactivation, can also confer functional deterioration when the accumulation of misfolded proteins during stress exceeds the HSC-specific, low tolerance[44–46]. The observed increase in ribosome biogenesis in fetal Fancd2[–/–] HSPC also aligns with observations in FA patients, and speaks to the recent suggestion that both loss or gain of protein synthesis can constrain HSC function[7,47]. To protect expansion in the

**Fig. 5 | Ifnar1 haploinsufficiency rescues Fancd2$^{-/-}$ FL HSPC deficits. A** Schematic representation of timed mating and experimental overview created with BioRender.com released under a Creative Commons Attribution-NonCommercial-NoDerivs 4.0 International license. **B** Genetic deletion of one Ifnar1 allele normalizes endogenous MYC levels in Ifnar1$^{+/-}$ Fancd2$^{-/-}$ ($n$ = 5/3) relative to Ifnar1$^{+/-}$ Fancd2$^{+/+}$, and Ifnar1$^{+/-}$ Fancd2$^{+/-}$ ($n$ = 6/3) FL HSPCs. **C** Relative rates of protein synthesis by in-vivo OPP analysis in Ifnar1$^{+/-}$ Fancd2$^{+/+}$, Ifnar1$^{+/-}$ Fancd2$^{+/-}$ ($n$ = 17/5) and Ifnar1$^{+/-}$ Fancd2$^{-/-}$ ($n$ = 7/5) FL HSPCs and representative histogram of LT-HSCs. **D** Relative levels of TPE-MI MFI measuring unfolded proteins in Ifnar1$^{+/-}$ Fancd2$^{+/+}$, Ifnar1$^{+/-}$ Fancd2$^{+/-}$ ($n$ = 17/3) and Ifnar1$^{+/-}$ Fancd2$^{-/-}$ ($n$ = 8/3) FL HSPCs and representative histogram of LT-HSCs. **E** Loss of one Ifnar1 alleles restores Fancd2$^{-/-}$ LT-HSC numbers to Fancd2$^{+/+}$ levels in fetal livers from E14.5 embryos: Fancd2$^{+/+}$ Fancd2$^{+/-}$ ($n$ = 37/11), Fancd2$^{-/-}$ ($n$ = 22/11) ($P$ = 0.01), Fancd2$^{-/-}$ ($n$ = 22/11) vs Ifnar1$^{+/-}$ Fancd2$^{-/-}$ ($n$ = 17/10) ($P$ = 0.04), Ifnar1$^{+/-}$ Fancd2$^{+/+}$, Ifnar1$^{+/-}$ Fancd2$^{+/-}$ ($n$ = 33/10) and Ifnar1$^{+/-}$ Fancd2$^{-/-}$ ($n$ = 17/10). **F** Colony forming unit (CFU) formation by Fancd2$^{+/+}$ ($n$ = 7/3) or Fancd2$^{-/-}$ ($n$ = 6/3) whole fetal liver cells (60 K) were plated on complete methylcellulose medium, and colonies were quantified after 7 days (far left panel = 0.045), CFU formation by adult BM lineage negative cells by (5 K) of Fancd2$^{+/+}$ ($n$ = 3 mice), Fancd2$^{-/-}$ ($n$ = 3 mice) (left panel $P$ = 0.02. CFU formation by Ifnar1$^{+/-}$ Fancd2$^{+/+}$ ($n$ = 9/3), Ifnar1$^{+/-}$ Fancd2$^{-/-}$ ($n$ = 9/3) whole fetal liver cells (60 K) (right panel $P$ = 0.9),and CFU formation by adult BM lineage negative cells (5 K) of Ifnar1$^{+/-}$Fancd2$^{+/+}$ ($n$ = 3 mice)), Ifnar1$^{+/-}$Fancd2$^{-/-}$ ($n$ = 3 mice) (far right panel, $P$ = 0.71). **G** Graphical abstract (created with BioRender.com released under a Creative Commons Attribution-NonCommercial-NoDerivs 4.0 International license) of the proposed model with increased sensitivity of Fancd2$^{-/-}$ FL HSPCs sterile inflammation driving proteostasis disruption and limited fetal HSC pool expansion in Fanconi Anemia mice. Data are represented as mean ± SEM. In **B**, **C**, **D**, and **E** One-way ANOVA was considered for statistical analysis, and in **F** data are represented as mean ± SD, $t$-test was used for statistical analysis. *$P$ < 0.05, **$P$ < 0.01, ***$P$ < 0.001, ****$P$ < 0.0001 and ns: non-significant. All statistical analyses are two sided and were performed with GraphPad Prism7.0 software. Source data are provided as a Source Data file.

fetal liver, HSPCs typically require suppression of ER stress, a process supported by fetal-specific bile acids that serve as chaperones to enhance folding[31]. We took advantage of the latter observation to show that fetal Fancd2$^{-/-}$ HSPC are amenable to pharmacological rescue by external bile acid supplementation. The molecular chaperone TUDCA, serially administered to midgestational dams, reduces the unfolded protein burden and rescued fetal Fancd2$^{-/-}$ cell numbers.

With proteostatic deregulation as a proximal mechanism for fetal HSC expansion failure in FA, we wanted to identify a plausible source. Again, careful analysis of the fetal Fancd2$^{-/-}$ HSPC and stromal transcriptome provided clues, with broad enrichment in inflammatory pathway activation in excess of what is seen during expansion of HSPCs in WT fetuses[2,48–50]. IFN-α is a physiologic component of the inflammatory response in fetal HSPCs during mid- and late gestation and critical for the transition from fetal to adult hematopoiesis[4,51]. Conceptually, the premise that sterile inflammation plays a role in FA pathophysiology is supported by the known vulnerability of FA hematopoietic cells to experimental inflammatory stress. For example, chronic stimulation with poly(I:C), a synthetic TLR-3 ligand and potent activator of IFN-α, leads to a collapse of (adult) FA hematopoietic function[21,52]. Compelling, direct evidence for a role during development comes from our study of Fancd2$^{-/-}$ Ifnar1$^{+/-}$ crosses that maintain a lower level of Type I IFN activity[51,53]. Here, the dampened fetal inflammatory response in these fetuses (in the context of a FANCD2 deficiency) blunted MYC levels and mitigated proteostasis disruption, sufficient to restore fetal LT-HSCs numbers and colony forming fitness in both fetal liver and adult BM cells. By contrast, a complete loss of IFNAR1 signaling in the double knockout (Fancd2$^{-/-}$ Ifnar1$^{-/-}$) crosses severely compromised viability and does not rescue FA HSPCs, strikingly similar to the fetal outcomes when TGF-β signaling in Fancd2$^{-/-}$ littermates is abrogated, as recently reported[36]. Together, these studies further confirm the fetal requirement for baseline levels of tonic inflammation. Likewise, broad pharmacologic inhibition of MYC dramatically reduces survival in FA cells that depend on some level on MYC activity and potentially selective MYC target activation.

Our model has limitations. Sterile inflammation in the FL environment involves several mediator pathways and cellular sources and we cannot fully rank effects of monocytes and specialized macrophages (Kupffer cells) versus HPSC's as sources of inflammatory signals in the FL niche[54]. From a pathophysiological point of view, this may be secondary as LT-HSC are clearly the most vulnerable to inflammation and proteostasis disruption, and most amenable to pharmacologic rescue. It is possible that other active pathways contribute to FA HSPC losses in the fetal liver. Like other recent papers, this study did not aim to define the precise signals that connect ER protein accumulation with HSPC proliferation, but it clearly demonstrates that restoration of protein folding as a rescue mechanism is operative in FA[7]. Experimental systems outside the FA phenotype may ultimately be more suitable for this. While there is no known sex-specific FA phenotype, our approach in evaluating both male and female embryos precludes any determination of sex bias in these observations. Finally, Fancd2 is one of 23 genes involved in FA, and while Fancd2$^{-/-}$ mice constitute a paradigmatic model system, our observations should be validated in other genotypes.

Thus, our study provides direct evidence that proteostasis deregulation constrains HSPC expansion in FA, revealing for the first time the developmental context and a mechanism that limits fetal HSPC numbers in a murine model of FA. The aggregate data support a working model (Fig. 5G), whereby pathophysiologic replication stress associated with FANCD2 loss of function and tonic sterile inflammation can activate MYC in the fetal hematopoietic environment. This accelerates protein synthesis and overwhelms ER folding capacity. The events that we show are amenable to pharmacological rescue, unlike apoptotic attrition seen in FA adult BM. Without FANCD2, however, ER stress response and p-Chk1 are engaged, respectively, for the protection of fetal proteome and genome integrity. The rapid pace of post-natal exhaustion of hematopoietic function in FA patients would appear to be a natural consequence of the profound fetal FA HSC pool deficits described here. More broadly, the current work suggests a link by which inflammation and proteostasis shape the developing HSC pool. Our observations imply that both unfolded protein response (UPR) and proteostasis may constitute potential therapeutic targets for the rescue of HSC pool development and functional fitness in FA.

## Methods
### Animal husbandry, timed pregnancies, and transplantation studies
All animal experiments were approved by OHSU or CHOP Animal Care and Use Committees, respectively. Murine Fancd2 is highly homologous to human FANCD2 with 44 exons (Gene bank accession #: BC042619) Fancd2 knockout (Fancd2$^{-/-}$) mice (C57BL/6 strain, CD45.2 isotype) were generated by a gene-trap method in the laboratory of Alan D'Andrea (Dana-Farber Cancer Institute, Harvard Medical School, Boston, MA). These Fancd2$^{-/-}$ mice had a phenotype similar to that observed in a previous report (Houghtaling et al.[55]). Upon gross examination Fancd2$^{-/-}$ mice were consistently smaller than Fancd2$^{+/+}$ than littermates indicating the general developmental defects in Fancd2$^{-/-}$ mice. Fancd2$^{+/+}$ and Fancd2$^{-/-}$ fetuses were harvested from timed pregnancies (vaginal plug method) generated by crossing heterozygous Fancd2$^{+/-}$ female with Fancd2$^{+/-}$ male mice. C57/BL6 background, CD45.2 isotype Fancd2 KO mice were bred and used for experiments. Fancd2$^{-/-}$ embryos (E14.5) were utilized for fetal liver studies. Transplantation studies were carried out with B6.SJL-Ptprca Pepcb/BoyJ (CD45.1 isotype) recipients. Fetal livers were dissected

from pups and separated by mechanical disruption, filtration, and subsequent red blood cell lysis to get mononuclear cells. Bone marrow was harvested from femurs and tibias of 8–12-week-old Fancd2$^{+/+}$ and Fancd2$^{-/-}$ animals. Tissues were genotyped for zygosity using primers sets and thermocycler reaction settings listed in supplementary Table 2. For studies involving Ifnar1 KO animals (CD45.2, C57BL/6) were purchased from Jackson Laboratory (colony Stock No. 028288) and 7–8-week-old mice were utilized for timed pregnancies to generate Fancd2$^{-/-}$ Ifnar1$^{+/-}$ embryos (E14.5) by crossing Fancd2$^{+/-}$ Ifnar1$^{+/+}$ mice with Fancd2$^{+/-}$ Ifnar$^{-/-}$ mice. Mice were housed in a specific pathogen-free (SPF)-grade controlled environment. ALPHA-dri was used as the cage bedding material with the room temperature ranging from 23–27 °C, and humidity of 30–45%. All mice were maintained under a 12-h light/dark cycle and had free access to sterile water and food (5015 (Lab Diet, catalog 0001328, with a metabolizable energy value of 3.59 kcal/g) in wired feeders above the floor of the cage. Animals were euthanized by exposing them to $CO_2$ (carbon dioxide) followed by the physical method of cervical dislocation.

There was no a priori blinding of experimental cohorts at the time of analysis. The experiments described otherwise conformed to the ARRIVE guidelines for reporting animal research.

## Transplantation

Harvested and RBC-lysed E12.5 FL cells ($5 \times 10^5$) were injected via the tail vein in CD45.1 recipients that received 750 cGy using an X-ray irradiator (single dose) At 20 weeks from transplantation, animals were sacrificed, tissues analyzed, and secondary transplantation was performed with injection of $1 \times 10^6$ whole bone marrow (WBM) BM cells into 750 cGy irradiated CD45.1 secondary recipients. Peripheral blood from both primary and secondary recipients was analyzed for chimerism, using antibodies against Gr-1, Mac-1, B220, CD3e and DAPI, by FACS. For adult BM transplant experiments, recipients received $1.5 \times 10^6$ WT and Fancd2$^{-/-}$ cells at a 1:1 ratio, followed by IP injection of TUDCA in one cohort or an equal volume of 1× sterile PBS.

## Mouse embryonic fibroblast (MEF) isolation

E14.5 embryos were harvested from timed pregnancy from Fancd2$^{+/-}$ females to yield Fancd2$^{-/-}$ and Fancd2$^{+/+}$ embryos. Tissues were minced and trypsinized for 10 min at 37 °C for 10 min. This was followed by washing with complete media and passage of the cell suspension through the 70 μm cell strainers to remove the tissue debris. Cells were then seeded into 0.2% gelatin-coated 100 mm cell culture dishes with 10 ml media (DMEM, 10% FBS, 1% PS) and passaged at ratios of 1:3. When confluent, MEFs were either used directly or stored by cryopreservation in DMSO.

## Colony formation unit (CFU) assay

Harvested and RBC-lysed whole FL or lineage-depleted adult BM cells were counted using trypan blue stain and mixed with cytokine-supplemented mouse methylcellulose media (R&D Systems, HSC007). Triplicates were cultured in 3.5 cm dishes at 37 °C. After 7 days, colony numbers were scored under an inverted light microscope.

## Immunophenotyping

Single cell resuspended FL or BM mononuclear cells underwent red cell lysis treatment before staining with cKIT at 1:100 concentration (BioLegend 105819), SCA1 (eBioscience 25-5981-81), CD135 (BioLegend), CD48 (BioLegend 103422), and CD150 (BioLegend 115925) antibodies, as well as the lineage antibodies: B220 (BD 553090), GR1 (BioLegend 108408), CD3 (BD 555275), CD4 (BD 553653), CD5 (BD 553023), and TER119 (BD 553673) at the manufacturer's recommended concentrations. Cells were stained for 30 min on ice, protected from light. Blocking and washing buffer contained 2%FBS/ PBS. For viability, dead cell exclusion staining using DAPI (Thermo 62248, 1μg/ml) was

included at a concentration of 1 μg/ml. Flow cytometric analysis was performed using FACS Canto2 and LSR2 instruments (BD Biosciences). For intracellular staining, FL or BM mononuclear cells were stained with surface markers and fixed with 2% PFA for 15 minutes, permeabilized with 0.5% saponin and stained with anti-p53, anti-p53S15 and anti-Cdc7. Samples were acquired with a LSR2 (Becton-Dickinson) and data were analyzed using FlowJo (10.6.1) software to quantify mean fluorescent intensity (MFI). Reagent Supplementaary Table 1 for details.

## Immunofluorescence Microscopy

FACS-sorted cells ($5–500 \times 10^3$) were spun onto glass slides using a cytocentrifuge, followed by incubation with or without Cytoskeletal (CSK) buffer for 10 min at room temperature and fixed in 4% PFA. Permeabilization was performed by the addition of 0.5% Triton and blocking with 3% BSA/PBS at 37 °C for 30 minutes. Primary antibody staining was performed on parafilm at 37 °C or 30 min, and secondary antibodies were used with 1:1000 dilution at 37 °C for 30 min. For nuclear staining, DAPI was used at room temperature for 10 min. For coverslip mounting we used Fluoromount-G (Southern Biotech, 0100). Images were captured on a Core DV microscope (Olympus) and via LSR700 confocal microscope (Carl Zeiss). Images were processed and analyzed with Imaris software (Bitplane). See Supplementary Table 1 for reagent details. To detect ssDNA, we used FlowSight (Amnis) and analyzed with IDEAs software (Amnis).

## Quantitative RT-PCR analysis

RNA was isolated using the RNeasy mini and micro kits (QIAGEN). SuperScript IV VILO Master Mix (Invitrogen) was used for cDNA synthesis. For RT-PCR, PowerUp™ SYBR™ Green Master Mix was used from Applied Biosystems. Multiplex real-time PCR was performed with the MYC targets PCR array (QIAGEN) following the manufacturer's instructions using the Applied Biosystems ViiA™ 7 Real-Time PCR instrument.

## EdU/ BrdU cell cycle assay

We used a previously reported assay[22] with sequential reagent injection via the tail vein in E13.5 pregnant females with 1 mg of EdU, followed 2 h later by 2 mg of BrdU. After 30 min FLs were harvested and individually processed. Isolated FL mononuclear cells were stained with surface markers (CD150, CD48, c-Kit, Sca-1, Lin) and fixed in 2% paraformaldehyde (PFA) for 15 min, followed by permeabilization with 0.5% saponin and stained with anti-EdU-AF488. To stain with an anti-BrdU antibody, we treated with 20ug of DNAse in PBS (containing Ca2+, Mg2+) at 37 °C for 40 min before staining with BrdU-AF647 (Thermo, B35140). Analysis was performed with FACS on an LSR2 instrument (Becton-Dickinson).

## O-Propargyl-puromycin (OPP) assay

For in vivo analysis of rate of translation O-Propargyl-Puromycin (OP-Puro) (Tocris Bioscience; 1416561-90-4) 50 mg/kg body mass; pH 6.4–6.6 resuspended in PBS was injected intraperitoneally into pregnant females at E-14.5 days, 1-hour later mice were euthanized. Dam BM and fetal liver cells were collected as described. RBC lysis was performed with 1× RBC lysis buffer, followed by cell were wash with 1× Ca2+ and Mg2+ (PBS). Stained with combinations of surface lineage markers (CD3, CD4, CD5, B220, Gr-1, Ter119), as well as CD150, CD48, c-Kit, Sca-1, and CD135 (for detail see Table S1) for 30 min on ice, in the dark, followed by washes with 1× PBS and fixation with BD fixative buffer for 15 min in the dark on ice. After washing with 1× PBS, cells were permeabilized in 200 μl PBS supplemented with 3% fetal bovine serum (Sigma) and 0.1% saponin (Sigma) for 5 min at room temperature, protected from light, and then an azide-alkyne click chemistry reaction was performed at room temperature using the Click-iT cell

Reaction Buffer Kit. Azide was conjugated to Alexa Fluor 555 (AF555) (Invitrogen, C10642) at 5 µM final concentration for 30 min. Dam bone marrow cells served as unstained control, i.e., they were stained as described above; however, click-iT chemistry reaction was performed without AF555. Samples were acquired by LSR Fortessa and analyzed by FlowJo (10.6.1) software. Results were plotted as OPP median fluorescence intensity (OPP-MFI) using the GraphPad Prism 7.0 software.

## MYC protein expression studies

To analyze the levels of MYC expression in Fancd2$^{-/-}$ fetal livers, cells were harvested from E14.5 embryos, and the dam femur BM cells as control. RBC lysis was performed, and cells were stained for 30 minutes on ice with antibodies against cell surface markers (CD3, CD4, CD5, B220, Gr-1, Ter119, CD150, CD48, c-Kit, Sca-1, and CD135; details in Table S1). After staining, cells were washed twice in Ca2+ and Mg2+ free phosphate buffered saline (PBS), fixed, permeabilized, and blocked. Cells were then stained with a primary anti-MYC antibody (Cell Signaling Technologies, D84C12) on ice for 30 minutes, followed by PBS washing and staining with a secondary antibody (Invitrogen,12-4739-81) for 30 min on ice. Cells were washed twice with PBS, and samples were acquired on LSR-Fortessa instrument. Data was analyzed by FlowJo (10.6.1) software and results were plotted as median fluorescent intensity using GraphPad Prism 7.0 software.

## Tetraphenylethene maleimide (TPE-MI) assay

To determine unfolded protein levels, fetal liver cells and bone marrow cells were harvested from the E-14.5 embryos and tibia and femur of the dams, respectively, followed by staining with cell surface markers described above. After surface epitope staining, cells were washed twice in Ca2+ and Mg2+ free phosphate buffered saline (PBS), followed by Tetraphenylethene maleimide (TPE-MI; stock 2 mM in DMSO) diluted in PBS (50 mM final concentration). This was added to each sample, and samples were incubated at 37 °C for 45 minutes. Samples were washed in PBS and fixed with fixative buffer for 15 min on ice then samples were washed twice with PBS and acquired by flow-cytometry using an LSR-Fortessa instrument. We cultured lineage-depleted bone marrow cells with 1 µM of Thapsigargin (Med Chem Express, HY-13433) for 4 h in vitro as a positive control; unstained cells served as an experimental negative control.

## Proteasome activity assay

LSK cells were sorted from the total fetal liver and dams BM cells after RBC lysis and surface staining and sorted cells were plated in triplicates ($5 \times 10^3$ cells in 100 µl/well) from each sample into the 96 well plates and 100 µl of reconstituted proteasome -Glow chymotrypsin like cell-based assay substrate (Promega) were added to each well and contents were mixed properly by plate shaker for 2 min at 700 rpm. After 10 min of incubation at room temperature luminescence was measured by Tecan infinite-M200 multimode plate reader and results were plotted by GraphPad Prism 7.0 software.

## Proteostat assay

Fetal livers from WT and Fancd2$^{-/-}$ FL were isolated and cells were sorted for KSL immunophenotype (Lin$^-$, c-kit$^+$, Sca-1$^+$) on a FACS -Aria Fusion sorter. Sorted cells were then seeded onto positively charged glass slides, fixed, permeabilized and incubated with Enzo Proteostat Detection Reagent (Enzo Life Sciences, cat #: ENZ-51035-K100). A DAPI nuclear counterstain was applied. Slides were mounted on coverslips and imaged with a Nikon Eclipse 50i microscope, using Micro-Manager analysis software.

## TUDCA (Tauroursodeoxy-cholic acid) treatment

TUDCA (Sigma-Aldrich) was dissolved in sterile PBS and that was intraperitoneally (IP) injected (7.5 mg/kg) into pregnant mice serially from embryonic day 10.5–14.5 of pregnancy and E14.5 embryos were harvested one hour after TUDCA injections for TPE-MI analysis. For ex vivo culture Fancd2$^{-/-}$ fetal liver cells were cultured for 18 h in Stem Span media, supplemented with IL-6 (500 µg), IL-3 (500 µg) and SCF (2500 µg), in 50 ml Stem Span and 1% PS. For adult BM transplant experiments, recipient animals received daily serial TUDCA (IP) injections followed by two injections per week for 10 weeks at 7.5 mg/kg.

## Single-cell RNA sequencing

DNA was extracted from a small amount of tissue from each embryo harvested at E13.5 using the KAPA Mouse Genotyping Kit (Roche, KK7352). The genotype and sex of each embryo was then determined via end-point PCR with the Hot Start Taq Master Mix Kit (Qiagen, 203443) or qRT-PCR with Power SYBR™ Green PCR Master Mix (Applied Bioscience, 4367659), respectively (Supplementary Table 4). Fancd2$^{-/-}$ embryos were identified, along with sex-matched WT littermate controls where possible. Fetal livers were dissected and mechanically digested in 900 µL ice-cold PBS + 0.04% BSA (PBS+) via pipetting, before shaking at 220 rpm for 10 min at 37 °C in 0.3% collagenase I (StemCell Technologies, 07416). Samples were washed with 5 mL ice-cold PBS+ and centrifuged, followed by resuspension in 2 mL ice-cold freshly prepared 1× RBC lysis buffer (Invitrogen, 00-4300-54). They were incubated on ice for 1 min and pelleted again, followed by resuspension in 1 mL ice-cold PBS+, and left on ice until staining for FACS. All centrifugation was performed at $400 \times g$ for 5 min at 4 °C, and all cell pellet resuspension was performed using wide-bore pipette tips. To safeguard adequate representation of smaller cell populations in the dataset, we first selectively depleted FL cells of the predominant Ter119 expressing erythroid cells before submission for library construction and sequencing. Samples were centrifuged and resuspended in a 1:100 dilution of FITC anti-mouse TER-119 antibody (BioLegend, 116206) with PBS and incubated on ice for 20 minutes, protected from light. After washing with 1 mL ice-cold PBS+, samples were centrifuged and resuspended in 300 µL ice-cold PBS+ with 5 µL propidium iodide (PI) solution (Miltenyi Biotec, 130-093-233). After at least 5 minutes on ice, protected from light, samples were sorted for live Ter119-negative cells using a FACS-Jazz instrument, and a cell count with Trypan blue was performed to assess cell viability. Samples were suspended in PBS at a concentration of 1000 cells/µL and submitted to the Children's Hospital of Philadelphia Center for Applied Genomics for same-day library generation.

## Library generation and 10X genomics sequencing

For library preparation, standard procedure was followed as written in the Chromium Single Cell 3′ Reagent Kits User Guide (v3.1 Chemistry). Library QC was performed using the Agilent Tape Station (High Sensitivity D1000 Screen Tape, 5067-5584; Reagents, 5067-5585) and KAPA Library Quantification Kit (Roche, 07960336001), followed by sequencing on the Illumina NovaSeq 6000 using a S2 100 cycles flow cell v1.5.

## Sequencing data analysis

Raw data was processed with Cell Ranger to obtain a gene-cell read count matrix. Data were filtered using Seurat version 4.0.1 to only include: genes detected in at least 3 cells; cells with 200–6000 detected genes cells with reads from mitochondrial genes less than 5% of total reads. Cells from all libraries were pooled together using the Seurat standard integration procedure, and the top 2000 genes with the highest variance. Cells were clustered by applying the PCA method to normalized read counts. Cell clusters were visualized by UMAP and manually assigned to known cell types based on their expression level of marker genes. Read counts of cells from the same biological replicate and cell type were pooled together to generate a gene-sample read count matrix. DESeq2 was applied to all gene-sample matrixes to

analyze the differential gene expression between two genotypes. The adjusted fold changes calculated by DESeq2 were used to rank genes for gene set enrichment analysis.

## Resource availability

Further information/requests about resources involved in this study should be directed to and will be fulfilled by the Lead Contact, Peter Kurre (kurrep@chop.edu).

## Statistical analysis

All numerical results were expressed as mean ± SEM. As appropriate, two-tailed Student's $t$ tests, Welch test and one-way ANOVA were performed for statistical analyses. All analyses were performed with GraphPad Prism7.0 software. For GSEA analysis both NES and $P$ values are statistical outputs from the GSEA run themselves.

## Reporting summary

Further information on research design is available in the Nature Portfolio Reporting Summary linked to this article.

## Data availability

Source data are provided with this manuscript. Single-cell RNA seq data have been deposited in Gene Expression Omnibus (GEO) with accession number: GSE173908. We also re-analyzed the published human scRNA-seq data set deposited in gene Expression Omnibus (GEO) under the accession number GSE157591. Source data are provided with this paper.

## Code availability

This paper does not contain original code.

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

## Acknowledgements

We are grateful for support by the Department of Pediatrics at OHSU and wish to thank Dr. Markus Grompe, Dr. Devo Goldman, Dr. Sherif Abdelhamed and Dr. Qingshuo Zhang for support and guidance with select experiments. We appreciated the support by OHSU and CHOP flow cytometry-, sequencing-, and animal core facilities. We are indebted to Julie Schrey for assistance with tailvein injections. Select figure panels and the visual abstract were created using BioRender™. Figure 1E was "Created with BioRender.com and has been granted a license (Agreement number: DF26FQMOLG) for use in journal publications. Figure 2A was "Created with BioRender.com and has been granted a license (Agreement number: FL26FQJLG7) for use in journal publications. Figure 3A was "Created with BioRender.com and has been granted a license (Agreement number: CY26FQNP36) for use in journal publications. Figure 3G was "Created with BioRender.com and has been granted a license (Agreement number: BC26FRFZQ6) for use in journal publications. Figure 4D was "Created with BioRender.com and has been granted a license (Agreement number: WA26FQPD16) for use in journal publications. Figure 5A was "Created with BioRender.com and has been granted a license (Agreement number: XZ26FQQUWN) for use in journal publications. Figure 5G was "Created with BioRender.com and has been granted a license (Agreement number: WA26FQR2XI) for use in journal publications. Supplementary Fig. 4C schematic was "Created with BioRender.com and has been granted a license (Agreement number: DH26FR34CF) for use in journal publications. We acknowledge helpful discussions and suggestions by lab members, including Dr. Stephanie Hurwitz and Suying Liu. Funding R01-HL150882 (PK).

## Author contributions

N.K., M.M.-K., T.M., and G.J. designed and performed research, wrote, and edited the manuscript; Y.H. contributed reagents and analytical tools and edited the manuscript; Y.M.Y. designed and performed experiments; Z.Z. led bioinformatics studies. P.K. designed research, wrote, and edited the manuscript.

## Competing interests

The authors declare no competing interests.

## Additional information

[1]Comprehensive Bone Marrow Failure Center, Children's Hospital of Philadelphia; Perelman School of Medicine, University of Pennsylvania, Philadelphia,
PA, USA. [2]Department of Microscopic and Developmental Anatomy, Tokyo Women's Medical University, Tokyo, Japan. [3]La Trobe University, Department of
Biochemistry and Chemistry, Melbourne, Australia. [4]Committee on Immunology, Graduate Program in Biosciences, University of Chicago, Chicago, IL, USA.
[5]Department of Biomedical and Health Informatics, Children's Hospital of Philadelphia, Philadelphia, PA, USA. [6]These authors contributed equally: Narasaiah
Kovuru, Makiko Mochizuki-Kashio. ✉e-mail: kurrep@chop.edu

