## [Peer Review File · Nature Communications]

REVIEWER COMMENTS

Reviewer #1 (Remarks to the Author):

The study by Kovuru et al provides an in-depth evaluation of the impacts on fetal HSC in *Fancd2* model of Fanconi Anemia. Although there are a series of typos (listed below), the overall text is clearly presented. This study presents compelling and convincing data the fetal HSCs in FA have dysregulated proteostasis using a variety of chemical and functional tools. Overall, this is a very strong study that will be impactful for the field. There are additional experimental studies that would further strengthen the study and model that are recommended if technically feasible.

The data presented in Figure 2 is used to support the claim that “*Fancd2*^{-/-} FL-HSC experience Myc activation? There is some support for this statement from the GSEA in Figure 1 but there is no functional or orthogonal validation of Myc activation presented in Figure 2. Authors should evaluate Myc levels and activation of known Myc targets in their elegant IF studies. It is possible that these data are shown in FigS1B but that data doesn't appear to be referenced in the main text. Same for FigS1C. This is further revealed in Figure 4 but the flow should be improved.

A final layer of confirmation of the model involving the genetic or pharmacologic inhibition of Myc would complete this study.

If technically feasible, an add back of *FANCD2* to correct the effect of the knockout, perhaps in the MEF model from 4D, would be powerful.

Please include a brief description of the *Fancd2* knockout to better understand the strain.

Figure 2, panel A. It is not clear how the data represented in the flow plots in panel A (right) compare to the data in panels B and C. The percent of EdU/BrdU double positive cells are 85% of the total with only 15% in the BM and these differences are more profound than those shown in panels B and C.

The authors show that TUDCA treatment (Figure 3H) corrects embryo weight but this is the first time that they mention embryo weight as a phenotype.

It is interesting that “Mono/DCs” appear to be the most perturb cell population in Figure 1H but the authors largely ignore this population.

There are some inconsistent uses of HSC and HSPC. Please standardize.

Some typos:

Extra period on lines 98, 157, 235, 834

Missing period on line 241

Random comma on line 236

Capital letter omission on line 236

Figure 3F, the labels for G, H and I are skewed.

Add legends for colors to 3G-3I.

Please change the language on line 203 as “MYC gains” is not appropriate as it implies copy number changes.

Reviewer #2 (Remarks to the Author):

Kovuru and colleagues present an interesting story linking FA-associated replication stress to inflammatory signaling, MYC, and disruption of proteostasis during the prenatal expansion of the hematopoietic compartment. The story is presented logically and is a nice overlap and extension of prior work from Dr. Kurre's group and from others. This is built on a strong premise established by the Signer and Magee groups. The experiments are well controlled and it is a strength that their analyses of fetal liver is compared to normal bone marrow as a reference. This markedly increases my confidence in their main findings. My comments are mainly focused on increasing rigor, impact and translational potential.

Main comments:

1) In the current version it is unclear how many independent experiments (or biologic versus technical replicates) went into each of the figure panels - this should be denoted in the figure legends for each panel, as should the statistical test used. In some panels (e.g., 2D, E, F, G) the mean/median and error

bars cannot be seen. It is suggested that description of replicates and statistical analysis should be improved throughout to raise the level of rigor.

2) Can the researchers access primary patient FA HSPCs and determine if there are elevated protein synthesis rates and/or unfolded proteins? Could they re-analyze D'Andrea's single cell RNA-seq dataset (or other human FA RNA-seq datasets) to determine if these signatures are present in the human setting? Could they look at these metrics of proteostasis in human FA fibroblasts, lymphoblasts, or one of the induced pluripotent stem cell models? Determining if their main conclusions are applicable in the human setting would enhance the impact and translational potential of this work.

3) I think that the authors definitions of MPP2/3/4 are outside of the accepted mainstream. Please see PMID 34648848 for the current consensus. MPP2 is typically E-biased, MPP3 is myeloid biased, and MPP4 (Lin-kit+Sca1+Flk2+) is generally lymphoid biased. The authors should at least provide a reference for their MPP designations.

4) Comparing Figs 1B and 3H, there seems to be about an order of magnitude difference in LT-HSC content at E14.5; this raises some concern as to the reproducibility of this flow-based quantification. Perhaps a representative gating scheme for the various populations could be added to the supplement and the quantification could be re-visited (or could it be clarified if these panels are actually different time points/methods of analysis?)?

5) The outcomes of the proteostasis/inflammation-targeted interventions (TUCDA treatment or *Ifnar* heterozygosity) are phenotypic metrics of LT-HSC quantity. Is there any evidence that these interventions improve *Fancd2*^{-/-} HSPC functionality as measured by colony formation or performance in transplantation (as done in S1A)?

6) An interesting follow-up question would be whether the approach of improving proteostasis via the fetal TUCDA treatment improves the function of postnatal *Fancd2*^{-/-} HSPCs - I would not mandate this experiment but it might be interesting for the authors to discuss potential therapeutic/translational implications.

Minor:

- The methods list some experimental techniques (e.g., CFU) that do not seem to be employed in the current version. I think some of the genes in Table S3 were not tested. I recommend reviewing the methods to ensure that it aligns with this version.

- Lines 72-73: 'The events in FA HSC' this sentence is a bit confusing and should be revised.

- Is Fig. S3A in vivo or in culture?
- Some of the p-values in selected GSEA signatures are not significant (e.g., lines 208-209). These conclusions could be enhanced by analyzing independent datasets as suggested above. Also, p-values should be included in Fig. 4C.
- I think the callout in line 216 should be for S2D.
- S2E-F are confusing - I think that the two genotypes are not labeled in these trajectory analyses and so it is difficult to vet the authors' interpretation of these data.
- Please quantify Fig. 4E.
- Fig. S1B shows that there are no differences in MYC between the two genotypes in the postnatal adult setting - this does not contrast with the basal state in the fetal liver (Fig. 4G) and so the language should be revised.
- Fig. S3D - the small, diffusely abnormal liver is hard to see on this gross image; can a scale bar be added or an image of a dissected liver or histology?
- The authors routinely analyze 'HSPC' populations, particularly in the immunofluorescence experiments - what markers define this population?
- Fig. 1H - why are some terms bolded? This does not seem to correlate with the signatures described in the results narrative.
- Fig. 3E-I - I assume the the color schemes are the same as in the prior panels but it might make sense to include a key for these panels. This also applies to S3E-H.
- Fig. 3G - is the rightmost column from wild-type?
- As I am also a researcher who works with mouse embryos, it is difficult to control for sex considerations in these models, and this is an accepted limitation in the field.

Grant Rowe

Boston Children's Hospital

Dear reviewers,

We appreciated the comments and critiques of our manuscript # NCOMMS-23-28394 (*Deregulated Protein Homeostasis Constrains the Fetal HSC Pool Expansion in Fanconi Anemia*), as well as the opportunity to address them experimentally.

While the availability of *Fancd2* KO embryos at a sub-mendelian frequency created an obvious bottleneck and accounts for the time to prepare this revision, we are pleased to provide a responsively revised manuscript and detailed rebuttal.

All changes in the attached revised manuscript are highlighted in **BLUE**, with point-by-point responses to specific questions -also in **BLUE**- below. To accommodate the additional experiments and figures, some panels have been moved and line references by reviewers to the original version changed accordingly.

SPECIFIC REVIEWER COMMENTS AND RESPONSES

Reviewer #1

General: The study by Kovuru et al provides an in-depth evaluation of the impacts on fetal HSC in *Fancd2* model of Fanconi Anemia. Although there are a series of typos (listed below), the overall text is clearly presented. This study presents compelling and convincing data the fetal HSCs in FA have dysregulated proteostasis using a variety of chemical and functional tools. Overall, this is a very strong study that will be impactful for the field. There are additional experimental studies that would further strengthen the study and model that are recommended if technically feasible.

Main comments:

- 1) The data presented in Figure 2 is used to support the claim that “*Fancd2*^{-/-} FL-HSC experience Myc activation? There is some support for this statement from the GSEA in Figure 1 but there is no functional or orthogonal validation of Myc activation presented in Figure 2. Authors should evaluate Myc levels and activation of known Myc targets in their elegant IF studies. It is possible that these data are shown in FigS1B but that data doesn’t appear to be referenced in the main text. Same for FigS1C. This is further revealed in Figure 4 but the flow should be improved.

Response: The manuscript lays out two central and related ideas, regarding dysregulated proteostasis and MYC activity, respectively, at the root of fetal hematopoietic failure in FA. We decided to present them in that sequence, first testing rescue of proteostasis, then inflammatory activation of MYC. Thus, while key data suggesting MYC involvement first emerge in Figure 1H and contribute to the discussion of Figure 2 (as the reviewer point out), the narrative flow of the manuscript, fully focuses on MYC beginning in Figure 4A. Here, we present MYC protein expression in *Fancd2*^{-/-} and WT HSPC populations, as suggested by the reviewer. The following Figure 5 revisits the *normalized* MYC protein levels in the *Ifnar1* *Fancd2*^{-/-} crosses. We also appreciated the reviewer’s query about additional data on MYC target activity given its central role as a transcription factor. We opted to address this by quantitative array profiling of key MYC targets. Collecting RNA from sorted KSL cells of *Fancd2*^{-/-} (n=11) and *Fancd2*^{+/+} (n=12) embryo-fetal livers, we used a multiplex RT2 profiler array. The results, now shown in **new Supplementary Fig. S5B**, confirm the broad activation of canonical MYC targets, including ribosome components, unfolded protein response elements, and cell cycle checkpoint genes, all speaking to the significance of aberrant MYC activity in the fate of fetal *Fancd2*^{-/-} HSPCs.

- 2) A final layer of confirmation of the model involving MYC’s genetic or pharmacologic inhibition would complete this study.

Response: The MYC pathway is critical for cell survival, proliferation, and HSC self-renewal, making *in vivo* studies of inhibition potentially challenging. Rodriguez et al (reference 21 within) showed that MYC signaling is critical for

sustaining adult FA HSPC activity in particular. (The authors propose a model whereby MYC reliance presents an alternate fate to the p53-mediated apoptotic fate that leads to bone marrow failure in FA bone marrow HSPC.) Responsive to the reviewer request, we took advantage of the (+)-JQ1 reagent (used by the authors in reference 21), a BRD4 inhibitor with a super-enhancer inhibitor, known to suppress MYC transcription. First, we injected pregnant mice with serial JQ1 doses (IP 50mg/KG; n=3 times) on E11.5, -12.5 and -13.5. This approach led to severe developmental defects and gross malformation of embryos apparent at the E14.5 harvest. On visual inspection JQ1 severely compromised the global liver development of embryos, which prevented any further recovery of HSPC and analysis in these embryos. We repeated this experiment with dose and schedule adjustments twice more, unable to recover sufficient cells for meaningful analysis in fetuses. Control staining in the bone marrow from the dam showed rates of protein synthesis (OPP) as a measure of MYC activity were comparable to historical BM HSPC controls, i.e. not indicative of MYC inhibition. The latter finding was in hindsight not unexpected, as Rodriguez and colleagues injected their animals for 30 consecutive days with JQ1. We conclude that *in vivo* injection of dams for fetal HSPC analysis was not feasible.

Next, we attempted *in vitro* JQ1 treatment to check whether MYC inhibition alleviates FA proteostasis defects in *Fancd2*^{-/-} BM LT-HSCs, *ex vivo* expanded using a validated polyvinyl-ETOH supplemented format. As shown previously by others (reference 33 within) *ex vivo* culture of BM HSPC is a potent inducer of proteostasis. First, we confirmed in our hands that less proliferative *fresh* adult BM cells do not have elevated levels of MYC and no significant increase in protein synthesis now updated as Supplementary Fig. S5C and S5D), whereas under *ex vivo* culture conditions *Fancd2*^{-/-} BM HSPCs have proliferative constraints and elevated unfolded proteins now shown in **Supplementary Fig. S4A and S4B**, respectively. However, when we used this format to test MYC inhibition by JQ1, we encountered a profound loss of HSPC viability and progenitor growth in both *Fancd2*^{+/+} and -more severely yet- in *Fancd2*^{-/-} conditions, shown in **new Supplementary Fig. S5F**.

Our aggregate conclusion from these studies is that MYC is a critical component in embryonic development and in maintaining *Fancd2*^{-/-} HSPCs. This aligns with dramatic losses in progenitor formation (Figure 4B, -F, and -G in reference 21) seen by Rodriguez et al., who proposed a model of “MYC reliance” of *Fancd2*^{-/-} BM HSPC.

- 3) If technically feasible, the add-back of FANCD2 to correct the effect of the knockout, perhaps in the MEF model from 4D, would be decisive.

Response: We appreciate the suggestion of experimental complementation *Fancd2* function. Unfortunately, we do not have a suitable lentivector in hand. When we reached out to no fewer than 4 laboratories at other institutions, none had a lentivector bearing the mouse cDNA sequence. A single publication of a mouse *Fancd2* lentivector existed (PMID: 25505262), but this laboratory had closed when the PI retired in 2021. We were able to locate a senior post doc from that lab, who -unfortunately- did not have this reagent in her lab. We then located a retrovirus vector in an additional lab, but that turned out to carry cDNA sequence different from that in the database when we performed Sanger sequencing for validation. Thus, while we understand the request and went to considerable length to comply, we would hope that the reviewer agrees that the WT murine cells provide a suitable comparison for our study, wherein the data consistently support the conclusions as presented.

- 4) Please include a brief description of the *Fancd2* knockout to better understand the strain.

Response: A description of the *Fancd2* knockout mice has been added to the materials and methods section, lines 419-425

- 5) Figure 2, panel A. It is not clear how the data represented in the flow plots in panel A (right) compared to the data in panels B and C. The percent of EdU/BrdU double positive cells is 85% of the total with only 15% in the BM and these differences are more profound than those shown in panels B and C.

Response: We apologize for the confusion. In flow plots in panel 2A, we simply lay out the procedure and illustrate how the method works, to highlight the uniquely fetal, highly proliferative HSPC phenotype and the ability of the analysis to resolve these (a fetal adaptation of this protocol has not been published previously). In panels 2B and

2C we compare fetal liver samples, all of which are highly proliferative by nature, between the genotypes (*Fancd2*^{+/+} vs *Fancd2*^{-/-}). Specifically, in panel 2B we showed percentages of EdU⁺ and or BrdU⁺ fetal liver cells among Lin^{-ve}, LSK, and LT-HSC populations in *Fancd2*^{+/+} and *Fancd2*^{-/-}. In panel 2C we show the percentage of cells newly entering S phase (BrdU⁺ EdU⁻/ BrdU⁺ + EdU⁺) in *Fancd2*^{+/+} versus *Fancd2*^{-/-} fetal liver samples. To avoid confusion for the reader, we have added a panel label for clarification.

6) The authors show that TUDCA treatment (Figure 3H) corrects embryo weight but this is the first time that they mention embryo weight as a phenotype.

Response: The reviewer is correct in that we did not report genotype weights elsewhere in the paper, even though we routinely record these details. We routinely found weight reduction in *Fancc*^{-/-} and *Fancd2*^{-/-} embryos in our two prior studies (references 13 and 15 within), and have now added embryo weights here for *untreated Fancd2*^{-/-} and WT littermates, **Fig. 3F**.

7) It is interesting that “Mono/DCs” appear to be the most perturbed cell population in Figure 1H but the authors largely ignore this population.

Response: We agree with the reviewers' observation that "Mono/DCs" appear to comprise an active population in the *Fancd2* fetal livers. While this novel observation deserves future attention, we felt it was beyond the scope of this study where we wanted to focus on our specific hypotheses related to hematopoietic stem cells.

Minor comments:

There are some inconsistent uses of HSC and HSPC. Please standardize. Some typos:

Response: We apologize for our inconsistency and in the revised manuscript we addressed these concerns.

Extra period on lines 98, 157, 235, and 834 Missing period on line 241

Response: We appreciate the reviewers' detailed scrutiny and lines 98, 157, 235, 241, and 834 were revised in the current version.

Random comma on line 236 Capital letter omission on line 236

Response: We addressed these issues in the revised manuscript.

Figure 3F, the labels for G, H, and I are skewed. Add legends for colors to 3G-3I.

Response: We appreciate the reviewer's suggestion, and we addressed the labels and colors, respectively for 3F, -G, -H, and -I in the revised manuscript.

Please change the language on line 203 as “MYC gains” is not appropriate as it implies copy number changes.

Response: We agree with the reviewer's suggestion and revised the phrase, now line 240 of the manuscript.

Reviewer #2 (Remarks to the Author):

General: Kovuru and colleagues present an interesting story linking FA-associated replication stress to inflammatory signaling, MYC, and disruption of proteostasis during the prenatal expansion of the hematopoietic compartment. The story is presented logically and is a nice overlap and extension of prior work from Dr. Kurre's

group and others. This is built on a strong premise established by the Signer and Magee groups. The experiments are well controlled, and it is a strength that their analyses of the fetal liver are compared to normal bone marrow as a reference. This markedly increases my confidence in their main findings. My comments are mainly focused on increasing rigor, impact, and translational potential.

Main comments:

1) In the current version it is unclear how many independent experiments (or biological versus technical replicates) went into each of the figure panels - this should be denoted in the figure legends for each panel, as should the statistical test used. In some panels (e.g., 2D, E, F, G) the mean/median and error bars cannot be seen. It is suggested that the description of replicates and statistical analysis should be improved throughout to raise the level of rigor.

Response: We appreciate the comment and have revised the figure legends for clarity do denote technical vs biological replicates. Regarding Figure 2D, -E, -F, and -G, we have changed the color of the error bars. Now error bars are clearly visible, and the method of statistical analysis also is specifically indicated in the legend.

2) Can the researchers access primary patient FA HSPCs and determine if there are elevated protein synthesis rates and/or unfolded proteins? Could they re-analyze D'Andrea's single-cell RNA-seq dataset (or other human FA RNA-seq datasets) to determine if these signatures are present in the human setting? Could they look at these metrics of proteostasis in human FA fibroblasts, lymphoblasts, or one of the induced pluripotent stem cell models? Determining if their main conclusions are applicable in the human setting would enhance the impact and translational potential of this work.

Response: Validation in primary cells can provide important validation. Unfortunately, the study of developmental hematopoiesis in human makes the relevant samples nearly impossible to obtain, especially in sufficient number, genotype and quality (LT-HSC). Unable to access other (non-fetal) primary samples, we welcomed the suggestion to re-analyze the publicly available single-cell RNA-seq dataset (Rodriguez et al, ref 21 within; GSE157591). Strikingly FA HSPCs in that data set also reveal elements of a dysregulated proteostasis signature, with upregulation of pathways such as endoplasmic reticulum (ER) stress, heat shock proteins/chaperones, and endoplasmic reticulum-associated degradation (ERAD) markers; now shown as a heat map in a **new Supplementary Fig. S4D**.

In addition to a dysregulated proteostasis signature, the dataset (GSE157591) also shows increased expression levels of the Interferon- α receptor-1 (IFNAR1) Interferon- α receptor-2 (IFNAR2), and Interferon- γ receptor-2 (IFNGR2) in FA patients HSPCs, consistent with the notion that elevated sterile inflammatory signaling is associated with FA pathogenesis, and aligning well with our results, **new Supplementary Fig. S4E**.

The reviewer's suggestion regarding the analysis of proteostasis disruption in induced pluripotent stem cell models is certainly interesting and timely, but we believe beyond the scope of this manuscript.

3) I think that the author's definitions of MPP2/3/4 are outside of the accepted mainstream. Please see PMID34648848 for the current consensus. MPP2 is typically E-biased, MPP3 is myeloid-biased, and MPP4 (Lin-kit+Sca1+Flk2+) is generally lymphoid-biased. The authors should at least provide a reference for their MPP designations.

Response: We appreciate this comment. To avoid any confusion, we have provided representative flow cytometry gating plots for HSPC subsets with identification with designated markers in the **new Supplementary Fig. S1A**. Additionally, we cited the relevant reference for our strategy (Challen *et al.*, Exp. Hematology 2021, now # 17) in the revised manuscript.

4) Comparing Figs 1B and 3H, there seems to be about an order of magnitude difference in LT-HSC content at E14.5; this raises some concern as to the reproducibility of this flow-based quantification. Perhaps a representative gating scheme for the various populations could be added to the supplement and the quantification could be revisited (or could it be clarified if these panels are different time points/methods of analysis?)

Response: We appreciate the comment regarding of differences in absolute cell numbers reported in Figs -1B and -3H (right panel). A set of representative flow plots with the gating scheme for the various populations under the different experimental conditions was added as a **new Supplementary Fig. 5A**.

To explain the difference in magnitude between Figure panels: For the quantification of LT HSCs numbers in Fig. 1B, live fetal liver cells were surface-stained immediately following harvest, in experiments dedicated to the immunophenotypic profiling and cell number quantification by flow cytometry. By contrast, LT-HSC numbers in Fig. 3H were derived from fetal liver samples processed for analysis of unfolded proteins (TPE-MI assay) and protein synthesis (OPP assay) using flow cytometry. Sample division for the different experiments and the processing that includes fixation formaldehyde) and permeabilization of the cells with 0.1 percent saponin contribute to lower numbers. Likely contributory as well is sample acquisition by two different flow cytometry instruments and operators for these two experiments. However, in spite of these differences, the proportional differences between genotypes are consistently maintained.

5) The outcomes of the proteostasis/inflammation-targeted interventions (TUDCA treatment or *Ifnar* heterozygosity) are phenotypic metrics of LT-HSC quantity. Is there any evidence that these interventions improve *Fancd2*^{-/-} HSPC functionality as measured by colony formation or performance in transplantation (as done in S1A)?

Response: We recognize this as a highly relevant comment. An obvious constraint in using FL cells for transplantation studies is the need to irradiate recipient mice 24 hrs in advance for transplantation (and thereby embryo harvest), without knowing if pregnancies yielded sufficient and appropriate genotype distribution among embryo litters. Therefore, to determine the changes in repopulation capacity of HSCs after proteostasis intervention with TUDCA, conducted transplantation experiments using adult *Fancd2*^{-/-} BM cells. First, we confirmed that *ex vivo* expanded HSPCs can mimic the proliferation constraints and proteostasis defects in FA (**new Supplementary Fig. S4A and S4B**). Here, results using-adult *Fancd2*^{-/-} BM cells for transplantation studies showed that *Fancd2*^{-/-} cells in TUDCA supplemented recipients transiently improve peripheral blood chimerism over *Fancd2*^{-/-} BM cells from PBS received mice. Predictably, because, TUDCA serves as a pharmacological protein folding chaperone, the continued supplementation would be required for long term, and more substantial, benefits. Thus, we provide proof of principle data, with long-term observations beyond the scope of this manuscript. These data are now shown in a **new Supplementary Fig. S4C**.

Next, to address the reviewer request for functional data on the impact of inflammation-targeted interventions by removing an *Ifnar1* allele, we determined colony forming capacity of the *Ifnar*^{+/-}/*Fancd2*^{-/-} fetal liver and BM cells compared with *Ifnar*^{+/-}/*Fancd2*^{+/+} progenitors. Results confirm that removal of the *Ifnar1* allele functionally improves the colony-forming ability of *Fancd2*^{-/-} fetal liver as well as -BM cells, these data are now added to **Fig.5F**.

6) An interesting follow-up question would be whether the approach of improving proteostasis via the fetal TUDCA treatment improves the function of postnatal *Fancd2*^{-/-} HSPCs

Response: It is an intriguing question. We would argue that this is quite likely given the effect in conditioning FA donors with TUDCA before transplantation of HSPC to irradiated recipients above. However, given sub-mendelian yield for breeding KO embryos, and the time to generate these animals, we were unable to generate these data.

7) interesting for the authors to discuss potential therapeutic/translational implications.

Response: We appreciate the reviewer's consideration of long-term implications in FA. To be appropriately cautious not to overinterpret our finding but added a responsive comment on the therapeutic implications of this study at

the end of the discussion section.

Minor comments:

The methods list some experimental techniques (e.g., CFU) that do not seem to be employed in the current version. I think some of the genes in Table S3 were not tested. I recommend reviewing the methods to ensure that they align with this version.

Response: We apologize for the discrepancies in the revised manuscript we ensured that they align with the updated version.

Lines 72-73: 'The events in FA HSC' This sentence is a bit confusing and should be revised.

Response: The sentence has been appropriately edited in the revised manuscript.

Is Fig. S3A in vivo or in culture?

Response: We regret the confusion. To validate the TPE-MI for unfolded protein analysis, we treated BM lineage-negative cells with Thapsigargin (1 μ M) for 4 hours under in-vitro culture conditions, now explicitly mentioned in the revised manuscript (lines 543-545).

Some of the p-values in selected GSEA signatures are not significant (e.g., lines 208-209). These conclusions could be enhanced by analyzing independent datasets as suggested above. Also, p-values should be included in Fig. 4C.

Response: We appreciate this comment and have included the p-values in each panel of Fig. 4C. To further substantiate our observations, we have revisited the publicly available dataset in FA patients HSPCs (GSE157591), and observed significantly increased expression levels of the IFNAR1, IFNAR2, and IFNGR2, now shown in **new Supplementary Fig. S4E**, and consistent with our GSEA inflammatory signaling signature (Fig. 1H).

I think the callout in line 216 should be for S2D.

Response: We apologize for the confusion caused and this has been addressed. Now in line 254 of the revised manuscript.

S2E-F are confusing - I think that the two genotypes are not labeled in these trajectory analyses and so it is difficult to vet the authors' interpretation of these data.

Response: We acknowledge the reviewer's concern regarding (S2E-F). Because lineage trajectory analysis in *Fancd2*^{-/-} mesenchymal cells and hepatoblasts remained unchanged compared to corresponding *Fancd2*^{+/+} (WT) cells, we elected to show an overlay of both *Fancd2*^{+/+} and *Fancd2*^{-/-} mesenchymal cell and hepatoblast lineage trajectories in the same figure panel. This is now clarified in the legend.

Please quantify Fig. 4E.

Response: We appreciate this suggestion, we quantified pChk1 and updated Fig. 4E with the quantification in the revised manuscript.

Fig. S1B shows that there are no differences in MYC between the two genotypes in the postnatal adult seVng - this does not contrast with the basal state in the fetal liver (Fig. 4G) so the language should be revised.

Response: Prior figure S1B has been moved to S5C. The reviewer correctly points out that results in that figure mirror those for the MEF model in Fig. 4G. We have revised this section accordingly (lines 228 to 230).

Fig. S3D - The small, diffusely abnormal liver is hard to see on this gross image; can a scale bar be added or an image of a dissected liver or histology?

Response: We acknowledge this comment and scale bars were added to the images in **Supplementary Fig. S3D**.

Regarding the fetal liver dissections and histology studies, we understand the reviewer's suggestion, but we have unfortunately not collected tissues for these analyses, and for the obvious reasons did not continue breeding or harvesting these genotypes.

The authors routinely analyze 'HSPC' populations, particularly in the immunofluorescence experiments - what markers define this population?

Response: We regret the confusion and, in our analysis, HSPC is defined as Lin⁻/Sca-1⁺/c-Kit⁺ cells. Related, we have carefully scrutinized our use of "HSPC" versus "LT-HSC" throughout the manuscript.

Fig. 1H - Why are some terms bolded? This does not seem to correlate with the signatures described in the results narrative.

Response: We understand the reviewer's comment, In Fig. 1H of GSEA analysis we initially bolded all the processes related to protein synthesis such as translation, ribosome biogenesis, t-RNA processes, and unfolded protein response including the mitochondrial ribosome biogenesis and mitochondrial translation to illustrate protein homeostasis defect in *Fancd2*^{-/-} fetal liver cells. The revised manuscript provides consistency between the terms bolded in **Fig. 1H** and the text narrative.

Fig. 3E-I - I assume the color schemes are the same as in the prior panels but it might make sense to include a key for these panels. This also applies to S3E-H.

Response: We agree with the reviewer, and we revised Fig. 3E-I and supplementary Fig. S3E-H with the key in the revised version.

Fig. 3G - is the rightmost column from wild-type?

Response: This rightmost column in Fig. 3G, now updated as **Fig. 3I** in the revised manuscript, analyzes unfolded proteins in bone marrow cells from the *Fancd2*^{+/-} dam. We have clarified this in the figure and legend.

As I am also a researcher who works with mouse embryos, it is difficult to control for sex considerations in these models, and this is an accepted limitation in the field.

Response: we agree with the reviewer it is difficult to control for sex considerations in these models. We added a sentence to this effect to the discussion.

REVIEWERS' COMMENTS

Reviewer #1 (Remarks to the Author):

The authors have sufficiently addressed my comments. The manuscript is improved and will be impactful for the field.

Reviewer #2 (Remarks to the Author):

The authors have performed a considerable amount of additional analysis. I only have a few minor comments to optimize the final version.

Line 72-73, suggest revising this new sentence to '...progression through DNA damage-induced cell cycle arrest in HSCs'. Progression through p53-mediated apoptosis would be associated with the cancer phenotypes.

Line 208 - callout should be to S4C.

Line 211 - callout should be to S4D-E.

Figure S2F - I still cannot see the differences in trajectory between the genotypes. The authors mention that there is actually a differences in the trajectory of hepatoblasts toward hepatocytes and reduced cholangiocytes in *Fancd2*^{-/-}. I understand that this is not a central conclusion but the interpretation is not apparent in looking at the panel since the genotypes are not distinguished. I would suggest including separate UMAPs for each genotypes or if this cannot readily be done removing these panels as their conclusions are not central.

Lines 311-12 - callouts should be to 5F.

Line 315 - callout should be to S5E.

Congratulations to the authors on this work which will hopefully positively impact FA patients in the future.

Grant Rowe

Dear reviewers,

We appreciated the comments and suggestions to further improve our manuscript # NCOMMS-23-28394A (Deregulated Protein Homeostasis Constrains the Fetal HSC Pool Expansion in Fanconi Anemia), as well as the opportunity to improve the manuscript presentation.

All changes in the attached revised manuscript are highlighted in **BLUE**, with point-by-point responses to specific questions -also in **BLUE**- below.

SPECIFIC REVIEWER COMMENTS AND RESPONSES

Reviewer #1

General:

The authors have sufficiently addressed my comments. The manuscript is improved and will be impactful for the field.

Reviewer #2

General:

The authors have performed a considerable amount of additional analysis. I only have a few monitor comments to optimize the final version.

Specific comments:

- 1) Line 72-73, suggest revising this new sentence to '...progression through DNA damage-induced cell cycle arrest in HSCs'. Progression through p53-mediated apoptosis would be associated with the cancer phenotypes.

Response: We appreciate the suggestion and lines 72-73 have been revised in the current version of the manuscript as per the reviewer's suggestion.

- 2) Line 208 - callout should be to S4C.

Response: We appreciate the reviewer's detailed scrutiny and line 208 has been revised appropriately. Now in line 206.

- 3) Line 211 - callout should be to S4D-E.

Response: We regret the discrepancies in line 211 and we revised the manuscript in text figure references appropriately. Now in lines 209-210.

- 4) Figure S2F - I still cannot see the differences in trajectory between the genotypes. The authors mention that there is a difference in the trajectory of hepatoblasts toward hepatocytes and reduced cholangiocytes in *Fancd2*^{-/-}. I understand that this is not a central conclusion, but the interpretation is not apparent in looking at the panel since the genotypes are not distinguished. I would suggest including separate UMAPs for each genotype or if this cannot readily be done removing these panels as their conclusions are not central.

Response: We appreciate the reviewer's point. To avoid confusion, separate independent t-SNE plots for each genotype WT and KO were added in revised Supplementary Fig. 2F.

5) Lines 311-12 - callouts should be to 5F.

Response: We appreciate the detailed scrutiny and the in-text Figure reference in lines 311-12 has been modified. Now referenced as Figure **5F** in lines 309- 310 of the current version.

6) Line 315 - callout should be to S5E.

Response: We apologize for the confusion. Now referenced as **S5E** in line 313 of the revised manuscript.

7) In line 576 you mention Table 1 which is not present in your manuscript, please check.

Response: We agree with the editor this manuscript doesn't contain any tables, we now updated that as a supplementary table 3 in line 578 of the revised manuscript and we confirm other supplementary tables were appropriately referenced in the current version.